# Late-life restoration of mitochondrial function reverses cardiac dysfunction in old mice

Ying Ann Chiao[1,2]*, Huiliang Zhang[1], Mariya Sweetwyne[1], Jeremy Whitson[1], Ying Sonia Ting[3†], Nathan Basisty[4], Lindsay K Pino[3], Ellen Quarles[1], Ngoc-Han Nguyen[1], Matthew D Campbell[5], Tong Zhang[6], Matthew J Gaffrey[6], Gennifer Merrihew[3], Lu Wang[7], Yongping Yue[8], Dongsheng Duan[8], Henk L Granzier[9], Hazel H Szeto[10], Wei-Jun Qian[6], David Marcinek[5], Michael J MacCoss[3], Peter Rabinovitch[1]*

[1]Department of Pathology, University of Washington, Seattle, United States; [2]Aging and Metabolism Program, Oklahoma Medical Research Foundation, Oklahoma City, United States; [3]Department of Genome Science, University of Washington, Seattle, United States; [4]Buck Institute for Research on Aging, Novato, United States; [5]Department of Radiology, University of Washington, Seattle, United States; [6]Biological Sciences Division, Pacific Northwest National Laboratory, Richland, United States; [7]Department of Environmental and Occupational Health Sciences, University of Washington, Seattle, United States; [8]Department of Molecular Microbiology and Immunology, School of Medicine, University of Missouri, Columbia, United States; [9]Department of Cellular and Molecular Medicine, University of Arizona, Tucson, United States; [10]Social Profit Network, Menlo Park, United States

*For correspondence:
Ann-Chiao@omrf.org (YAC);
petersr@uw.edu (PR)

†Deceased

Competing interests: The authors declare that no competing interests exist.

**Abstract** Diastolic dysfunction is a prominent feature of cardiac aging in both mice and humans. We show here that 8-week treatment of old mice with the mitochondrial targeted peptide SS-31 (elamipretide) can substantially reverse this deficit. SS-31 normalized the increase in proton leak and reduced mitochondrial ROS in cardiomyocytes from old mice, accompanied by reduced protein oxidation and a shift towards a more reduced protein thiol redox state in old hearts. Improved diastolic function was concordant with increased phosphorylation of cMyBP-C Ser282 but was independent of titin isoform shift. Late-life viral expression of mitochondrial-targeted catalase (mCAT) produced similar functional benefits in old mice and SS-31 did not improve cardiac function of old mCAT mice, implicating normalizing mitochondrial oxidative stress as an overlapping mechanism. These results demonstrate that pre-existing cardiac aging phenotypes can be reversed by targeting mitochondrial dysfunction and implicate mitochondrial energetics and redox signaling as therapeutic targets for cardiac aging.

## Introduction

Mitochondrial dysfunction is one of the hallmarks of aging (*López-Otín et al., 2013*). While mitochondria generate the bulk of cellular ATP, they are also the major source of reactive oxygen species (ROS) in most cells. The mitochondrial free radical theory of aging proposes that excessive mitochondrial ROS damages mitochondrial DNA and proteins, and this leads to further mitochondrial dysfunction, with subsequent cellular and organ functional declines and limits on lifespan and healthspan (*Harman, 1972*).

Aging is the strongest risk factor for cardiovascular diseases (*Niccoli and Partridge, 2012*). It is also accompanied by a decline in cardiac function, especially diastolic dysfunction and hypertrophy of the left ventricle and left atrium (*Lakatta and Levy, 2003*). The heart is rich in mitochondria and has a high metabolic demand; therefore, it is highly susceptible to oxidative damage and the effects of mitochondrial dysfunction. Increasing evidence suggests that mitochondrial oxidative stress and mitochondrial dysfunction play critical roles in cardiovascular diseases and cardiac aging (*Tocchi et al., 2015*).

The therapeutic potential of reducing mitochondrial oxidative stress is supported by mice expressing mitochondrial-targeted catalase (mCAT) (*Schriner et al., 2005*). In these mice, catalase removes hydrogen peroxide in mitochondria and significantly reduces mitochondrial protein oxidative damage and mitochondrial DNA mutation and deletion frequencies in mCAT mice. In addition to an extension of median and maximum lifespan, mCAT mice displayed greatly attenuated cardiac aging phenotypes, including reduced cardiac hypertrophy and improved diastolic function and myocardial performance (*Dai et al., 2009*). Expression of mCAT is also protective in models of cardiac hypertrophy and failure (*Dai et al., 2011a*). These cardiac benefits suggest that pharmacologic interventions combating mitochondrial ROS and improving mitochondrial function are attractive targets for treatment of cardiovascular disease and cardiac aging. Despite the positive effects of targeting mitochondrial ROS by mCAT on lifespan and in multiple disease models, studies have also reported negative effects of targeting mitochondrial ROS. While normalizing mitochondrial ROS by modest level of mCAT expression attenuates cardiac defects in a model of mitofusin-deficient cardiomyopathy, super-suppression of mitochondrial ROS by high level of mCAT expression exacerbates the defects (*Song et al., 2014*). In another study, suppression of mitochondrial ROS in mice resulted in impaired macrophage bactericidal activity (*West et al., 2011*). A recent proteomic study demonstrated that while old mCAT mice displayed a more youthful proteome composition and turnover compared to old wild-type mice, the proteome of young mCAT mice recapitulates that of old wild-type mice (*Basisty et al., 2016*). These studies support the physiological roles and an age-dependent pleiotropy of mitochondrial ROS (*Basisty et al., 2016*; *Sena and Chandel, 2012*). Therefore, later-life interventions that target pathological levels of mitochondrial ROS in old age may offer better translational potentials than life-long or long-term interventions.

We focused on the mitochondrial-targeted tetrapeptide SS-31 (elamipretide), a pharmacologic intervention that selectively concentrates in mitochondria, suppressing mitochondrial ROS (*Szeto, 2006*) and increasing skeletal muscle ATP production (*Campbell et al., 2019*; *Siegel et al., 2013*). SS-31 treatment was previously shown to reduce mitochondrial oxidative damage and prevent pressure overload-induced cardiac hypertrophy and failure in a manner that was highly similar to mCAT (*Dai et al., 2011b*; *Dai et al., 2013*; *Dai et al., 2012*). While these and other studies have shown that combating mitochondrial ROS during the course of a lifetime or during work and pressure overload stress can prevent mitochondrial dysfunction and attenuate cardiac functional decline (*Dai et al., 2014a*; *Dai et al., 2017*), it has not been established whether delivering such interventions in later life can rescue pre-existing mitochondrial and cardiac dysfunction. In this study, we demonstrate that mitochondrial-targeted interventions can improve mitochondrial function and reverse pre-existing cardiac dysfunction in old mice.

## Results

### 8-week SS-31 treatment rescues cardiac dysfunction and hypertrophy in old mice

Diastolic function and myocardial performance decline significantly with age (*Dai et al., 2009*; *Chiao et al., 2012*; *Dai et al., 2014b*). Compared to young mice, old mice exhibit a reduced ratio of early to late diastolic mitral annulus velocities (Ea/Aa), indicating a decline in diastolic function, and they have an increased (poorer) myocardial performance index (MPI), indicating an increased fraction of the cardiac cycle that is not accompanied by a change in volume (*Dai et al., 2009*; *Chiao et al., 2012*; *Dai et al., 2014b*). To determine the effects of SS-31 treatment on cardiac function in old mice, we treated 24-month-old mice with the SS-31 peptide or saline control and examined cardiac function by echocardiography after 4 and 8 weeks of treatment. We found that Ea/Aa increased and MPI decreased in old mice treated with SS-31 for 8 weeks, reversing the age-related changes, and

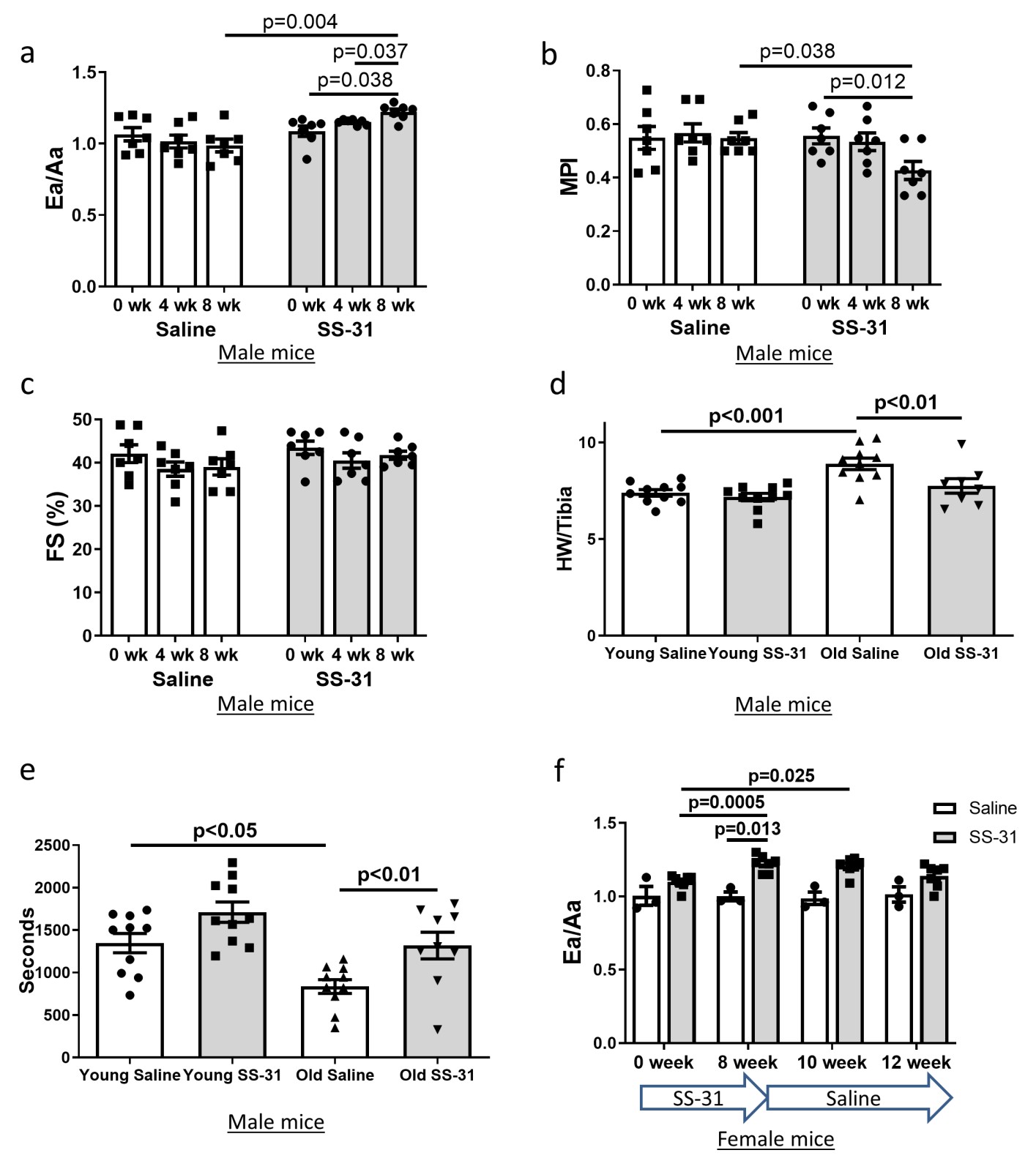

**Figure 1.** SS-31 treatment reverses cardiac aging phenotypes and improves exercise performance in old mice. Doppler echocardiography showed that 8-week SS-31 treatment (**a**) improved diastolic function (increased Ea/Aa) and (**b**) enhanced myocardial performance (reduced myocardial performance index, MPI) of old male mice. (**c**) Fractional shortening (FS) was not altered by SS-31 treatment. (**a–c**) n = 7 male mice/group were analyzed by repeated measure ANOVA with Tukey's multiple comparison test between time points and Sidak post hoc analysis between treatment groups. (**d**) 8- week SS-31

*Figure 1 continued on next page*

*Figure 1 continued*

treatment reversed the age-related increase in normalized heart weight. n = 10 (for young saline, young SS-31 and old saline) and n = 8 (for old SS-31) male mice. Data were analyzed by one-way ANOVA with SNK post hoc analysis. (e) Treadmill running was impaired (reduced running time) in old control male mice but was rescued by SS-31 treatment. n = 10 (for young saline, young SS-31 and old saline) and n = 9 (for old SS-31) male mice, analyzed by one-way ANOVA with SNK post hoc analysis. (f) The improved diastolic function in old female mice (increased Ea/Aa) after 8 week of SS-31 treatment persisted for 2–4 weeks after cessation of treatment. n = 3 for saline control and n = 7 for SS-31 treatment, analyzed by repeated measure ANOVA with Tukey's multiple comparison test between time points and Sidak post hoc analysis between treatment groups.

The online version of this article includes the following figure supplement(s) for figure 1:

**Figure supplement 1.** Individual trajectories of SS-31 responses in cardiac function of male mice.
**Figure supplement 2.** Individual trajectories of SS-31 responses in exercise performance of male mice.
**Figure supplement 3.** Individual trajectories of SS-31 responses in cardiac function of female mice.

both parameters were significantly different compared to saline controls at 8 weeks of treatment (*Figure 1a,b*, *Figure 1—figure supplement 1*). Systolic function, measured as fractional shortening, was not altered by SS-31 treatment and remained similar between old control and old SS-31 treated mice (*Figure 1c*, *Figure 1—figure supplement 1*). At the 8-week necropsy, we observed a higher heart weight normalized to tibia length (HW/TL) in old control mice compared to young control mice, while HW/TL of old SS-31 treated mice was lower than that of old controls (*Figure 1d*), suggesting a regression of age-related cardiac hypertrophy after SS-31 treatment. A decline in diastolic cardiac function in the elderly is associated with exercise intolerance, so we studied whether exercise performance was improved by SS-31 treatment. We observed reduced treadmill running time in old mice compared to young mice, and old mice treated with SS-31 for 8 weeks ran significantly longer than old control mice (*Figure 1e*, *Figure 1—figure supplement 2*), consistent with recent observations (*Campbell et al., 2019*). As in male mice, we observed a similar improvement in Ea/Aa in 24 month old female mice treated with SS-31 for 8 weeks (*Figure 1f*, *Figure 1—figure supplement 3*), suggesting that the treatment is effective in both sexes. To evaluate the persistence of the SS-31-induced cardiac benefit, we continued to monitor cardiac function in these mice after cessation of treatment. We found that the improved Ea/Aa in SS-31 treated mice was maintained at 2 weeks, but dropped by approximately half at 4 weeks after treatment ceased (*Figure 1f*, *Figure 1—figure supplement 3*).

## SS-31 treatment suppresses mitochondrial ROS production in old cardiomyocytes

The SS-31 peptide has been shown to attenuate mitochondrial oxidative stress in multiple disease models (*Dai et al., 2011b*; *Dai et al., 2014a*; *Tarantini et al., 2018*). To investigate its effect on mitochondrial ROS production in cardiomyocytes, we isolated cardiomyocytes from old control and old SS-31 treated mice and measured mitochondrial ROS production with fluorescent indicators of ROS. Confocal microscopy revealed reduced MitoSOX intensity in cardiomyocytes from old SS-31 treated mice, indicating reduced mitochondrial superoxide production (*Figure 2a*), as well as reduced MitoPY1 fluorescence, a measure of mitochondrial hydrogen peroxide production (*Figure 2b*).

## Increased mitochondrial proton leak in old cardiomyocytes is normalized by SS-31 treatment

To determine the effect of SS-31 treatment on mitochondrial respiration, we assessed the oxygen consumption rate (OCR) in isolated adult cardiomyocytes using the Seahorse Bioscience XF Cell Mito Stress Test assay. Basal respiration was significantly higher in cardiomyocytes from old control mice compared to cardiomyocytes from young mice, and this age-related increase in basal respiration was normalized in cardiomyocytes from old SS-31 treated mice (*Figure 2c,d*). These changes in basal respiration were almost entirely the result of altered proton leak, which increased in old cardiomyocytes and was normalized by SS-31 treatment (*Figure 2c,e*). In addition, the respiratory control ratio (RCR) decreased in old cardiomyocytes, and this was partially restored by SS-31 treatment. We also measured mitochondrial membrane potential in old cardiomyocytes treated with SS-31, as measured with the dye JC-1. We found increased membrane potential in cardiomyocytes from old SS-31 treated mice ( *Figure 2—figure supplement 1*), which is consistent with the observed decreased

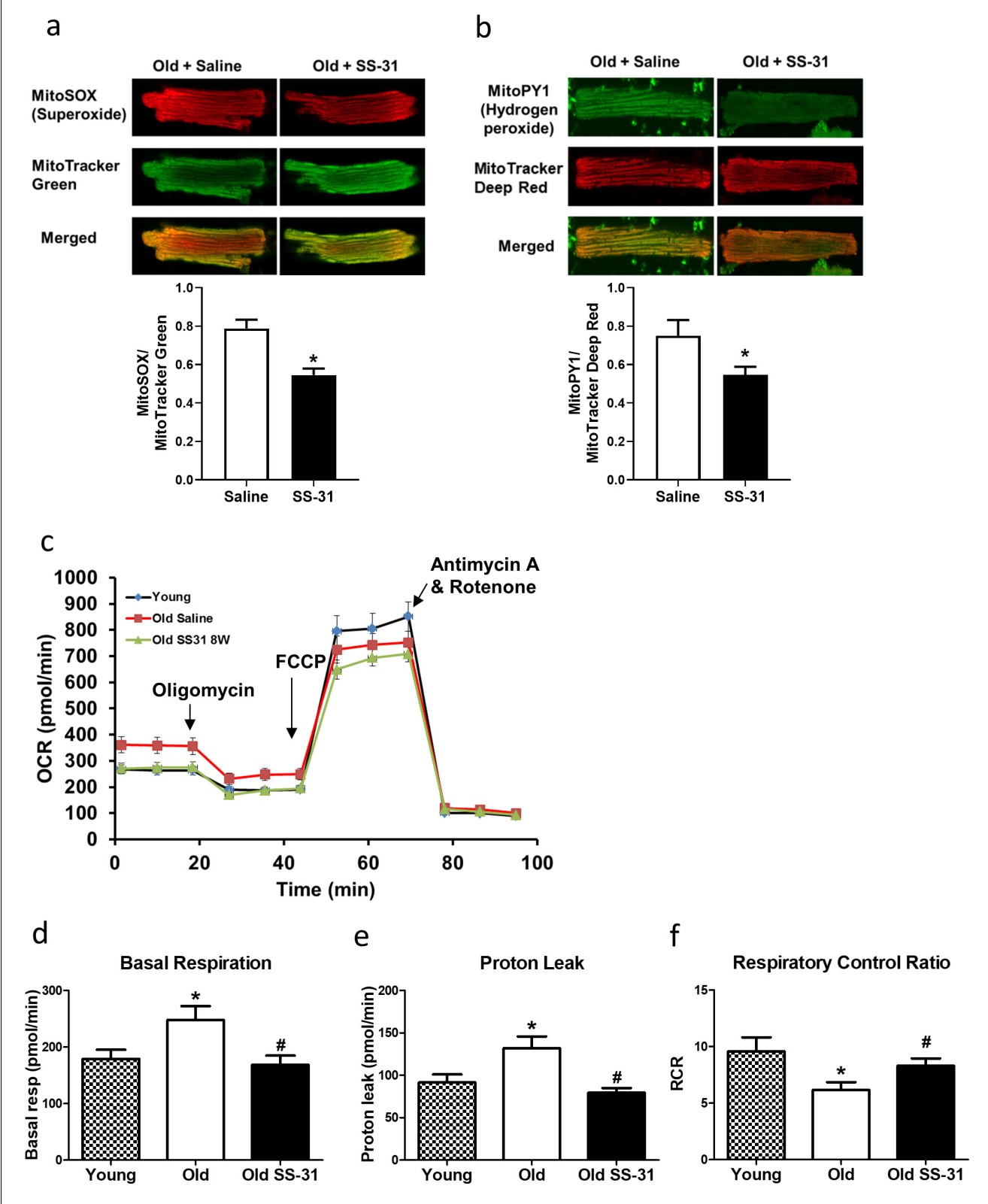

**Figure 2.** SS-31 treatment reduces ROS production and improves respiration in cardiomyocytes. (a) SS-31 treated cardiomyocytes showed reduced mitochondrial superoxide, indicated by reduced MitoSox signal (normalized to mitochondrial content by the ratio to MitoTracker Green), compared to old controls. *p<0.05 vs old saline; n = 67 cells from three female mice for old saline and n = 71 cells from three female mice for old SS-31, compared by unpaired T-test. (b) SS-31 treated cardiomyocytes showed reduced hydrogen peroxide, indicated by reduced mitoPY1 signal (normalized to

*Figure 2 continued on next page*

*Figure 2 continued*

mitochondrial content using MitoTracker Deep Red), compared to old controls. *p<0.05 vs old saline; n = 31 cells from three female mice for old saline and n = 29 cells from three female mice for old SS-31, compared by unpaired T-test. Images for MitoSox and MitoPY1 measurements can found in *Figure 2—source data 1* and *Figure 2—source data 2*. (c) Averaged traces of oxygen consumption rate (OCR, + / - SEM) of isolated cardiomyocytes from young, old, and old SS-31 treated male and female mice measured by the Seahorse XF Cell Mito Stress Test. Cardiomyocytes from old mice exhibited increased basal respiration (d) and proton leak (e) compared to that of young mice, and these age-related increases were reversed in cardiomyocytes from 8-week SS-31 treated old mice. (f) Old cardiomyocytes exhibited reduced respiratory control ratio (RCR) compared to young cardiomyocytes and this decrease was partially restored by 8-week SS-31 treatment. (d–f) *p<0.05 vs. young saline; #p<0.05 vs. old saline; n = 16 wells from four mice for young, n = 29 wells from six mice for old and n = 35 wells from six mice for old SS-31, analyzed by one-way ANOVA with SNK post hoc analysis.

The online version of this article includes the following source data and figure supplement(s) for figure 2:

**Source data 1.** All image files for MitoSOX analysis in *Figure 2a*.
**Source data 2.** All image files for MitoPY1 analysis in *Figure 2b*.
**Figure supplement 1.** SS-31 treatment increases mitochondrial membrane potential in aged cardiomyocytes.

proton leak. We tested whether the improved RCR and reduced proton leak in SS-31 were accompanied by changes in levels of oxidative phosphorylation (OXPHOS) complexes; however, we observed no change in abundance of subunits of OXPHOS complexes after 8-week SS-31 treatment (*Figure 3*).

## SS-31 treatment reduces protein oxidation and cellular senescence in old hearts

Mitochondrial oxidative stress can lead to oxidative modifications of cellular proteins. We studied whether the extent of Cys S-glutathionylation, an important reversible oxidative posttranslational modification in response to oxidative stress (*Shelton and Mieyal, 2008*), was affected by aging or SS-31 treatment. Overall, proteins in young control hearts have an average of 5.3% occupancy by glutathionylation; this increased by 33% to 7.1% occupancy in old control hearts, but 8-week SS-31 treatment reduced the glutathionylation occupancy of old heart proteins to 5.9%. At the individual peptide level, cardiac proteins from old control mice have increased levels of glutathionylation in the majority of detected peptides compared to young controls, indicating a general age-related increase in protein glutathionylation, substantially and broadly reduced by SS-31 treatment (*Figure 4a*). We also assessed levels of protein carbonylation, another protein oxidative modification, often viewed as a hallmark of oxidative damage (*Dalle-Donne et al., 2003*; *Fedorova et al., 2014*). We detected an increase in protein carbonylation in old control compare to young hearts, and this age-related increase was abolished by SS-31 treatment (*Figure 4b*).

Mitochondrial dysfunction can induce cellular senescence (*Wiley et al., 2016*). To determine if the improved mitochondrial respiration and reduced oxidative stress in old SS-31 treated mice was associated with reduced cellular senescence, we examined cellular senescence by immunostaining of senescent markers, p16 and p19, in hearts of old control and old SS-31 treated mice. And indeed, there were fewer senescence cells with p16-positive nuclei or p19-positive nuclei in the SS-31 treated old hearts (*Figure 4c and d*).

## SS-31 treatment partially restored aging-induced changes in the proteome and metabolome

We performed global proteomic analyses by mass spectrometry to study the changes in protein abundance induced by SS-31 treatment. We detected 277 proteins with altered expression levels with aging (q < 0.05 for old controls compared to young controls) and 192 proteins with altered levels in old mice after 8 weeks of SS-31 treatment (q < 0.05 for old SS-31 compared to old controls) . Expression levels of 88 proteins were significantly altered by both aging and SS-31 treatment, and SS-31 attenuated the aging-induced changes for a majority of these proteins (*Figure 5*). The Ingenuity Pathway Analysis (IPA) top canonical pathways affected by both aging and SS-31 included mitochondrial dysfunction, oxidative phosphorylation, GP6 signaling pathway and sirtuin signaling pathway (p<2.8E-06). However, in comparison with results previously reported for SS-31 treatment in hypertensive heart failure (*Dai et al., 2013*), these changes were much smaller in magnitude.

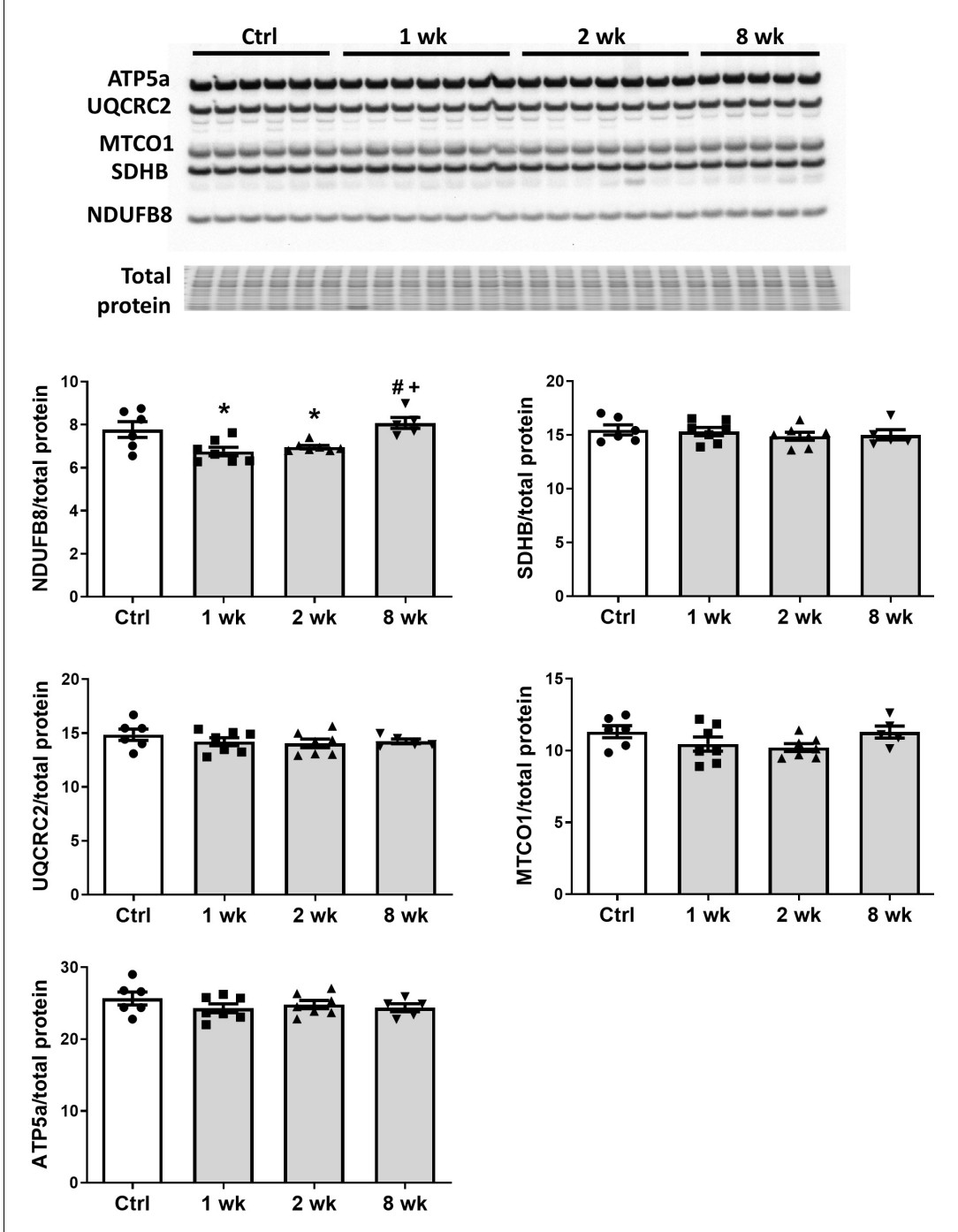

**Figure 3.** SS-31 treatment does not alter expression of subunits of oxidative phosphorylation complexes. Immunoblotting using anti-OXPHOS antibody detected no differences in expression levels of OXPHOS subunits (NDUFB8, SDHB, UQCRC2, MTCO1, and ATP5A) in hearts of old male mice treated with SS-31 for 8 weeks. Only transient changes in NDUFB8 levels were detected at 1 and 2 weeks after SS-31 treatment. *p<0.05 vs Control; +p<0.05 vs 1-week SS-31; #p<0.05 vs 2-week SS-31 treatment; n = 6 for Control, n = 7 for 1 week and 2 week and n = 5 for 8 week, analyzed by one-way ANOVA with SNK post hoc analysis.

We also performed targeted metabolic profiling on cardiac tissue from young and old mice with SS-31 or saline treatment. Out of the 160 metabolites measured, 112 metabolites were detected in all samples (*Supplementary file 1*) and the levels of 18 metabolites were significantly different among the groups (FDR < 0.05, *Supplementary file 2* and *Figure 5—figure supplement 1*). Age-

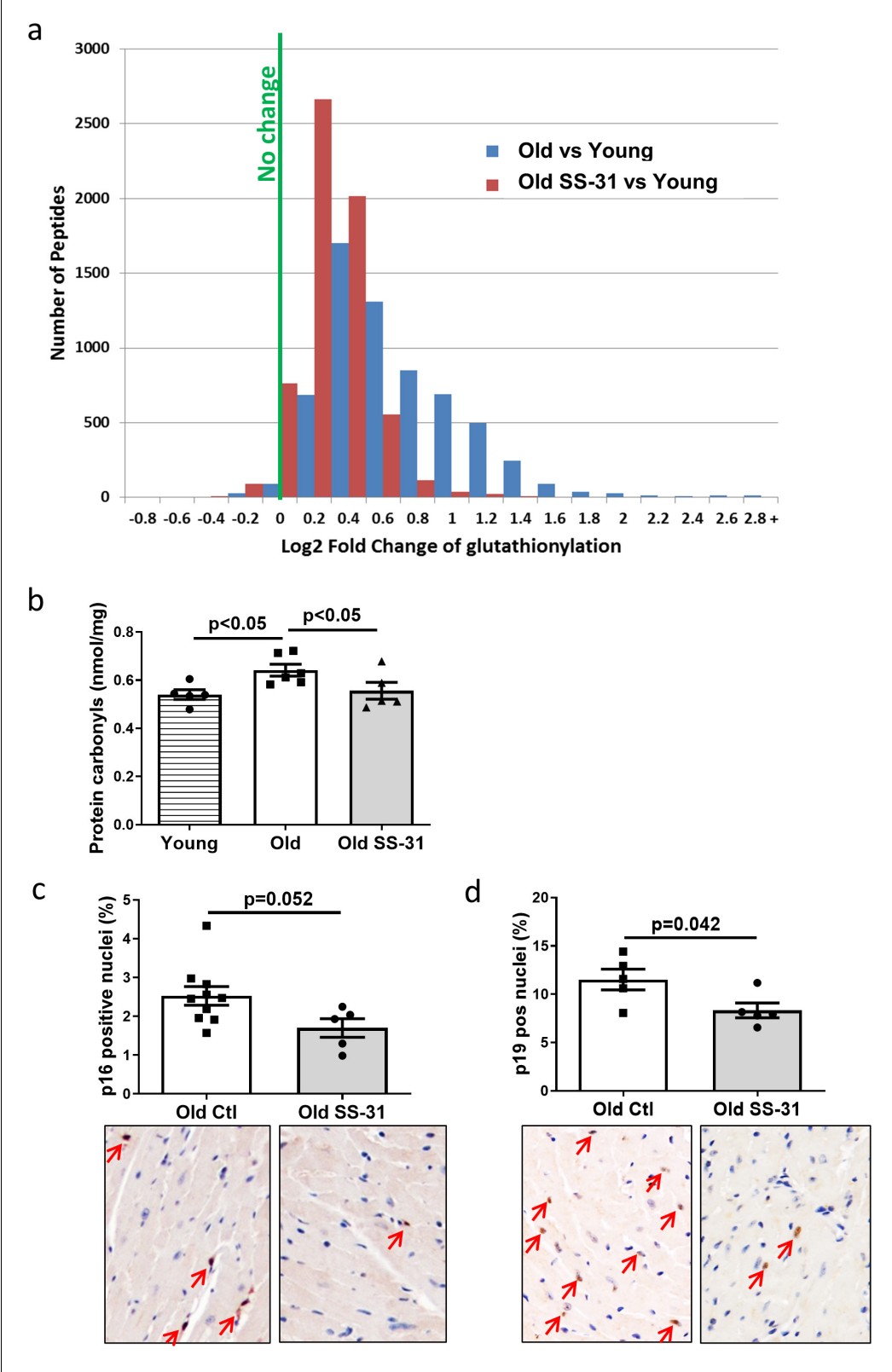

**Figure 4.** SS-31 treatment reduces protein oxidation and senescence in old hearts. (**a**) A histogram of the distribution of changes in glutathionylation levels in peptides from old control and old SS-31 treated hearts; n = 3 female mice per group, analyzed as described in the Materials and method section. (**b**) Increased levels of protein carbonylation were detected in hearts of old control mice, but not old SS-31 treated mice, when compared to young control mice; n = 5 female mice per group, analyzed by one-way ANOVA with SNK post hoc analysis. (**c–d**) IHC staining of cellular senescence

*Figure 4 continued on next page*

*Figure 4 continued*

markers, p16 (c) and p19 (d), detected reduced p16 and p19 positive nuclei in old SS-31 treated heart compared to old control hearts; n = 5 male mice per group, analyzed by unpaired T-test. Images for p16 and p19 staining can be found in *Figure 4—source data 1* and *Figure 4—source data 2*. The online version of this article includes the following source data for figure 4:

**Source data 1.** All image files for p16 analysis in *Figure 4c*.
**Source data 2.** All image files for p19 analysis in *Figure 4d*.

related reductions in metabolite levels were significant in 11 of the 18 metabolites and while none of these were significantly different between old control and old SS-31 groups, SS-31 partially attenuated these age-related metabolic changes (*Supplementary file 2* and *Figure 5—figure supplement 1*). Enrichment analysis was performed to gain biological insight into the age-related metabolic changes and revealed that two metabolite sets, aspartate metabolism and urea cycle, were significantly enriched (FDR < 0.05) in the 11 metabolites showing age-related changes. A network view of the Enrichment Analysis is shown in *Figure 5—figure supplement 2*.

## SS-31 treatment normalized age-related hypo-phosphorylation of cMyBP-C at Ser282

Myofilament proteins are important regulators of cardiac muscle contraction and relaxation. Phosphorylation of myofilament proteins modulates myofilament properties and regulates the relaxation behavior of cardiac muscle (*Biesiadecki et al., 2014*). More specifically, phosphorylation of cardiac myosin binding protein C (cMyBP-C) can modulate cross-bridge detachment and diastolic function (*Tong et al., 2014*). Old hearts displayed hypo-phosphorylation of MyBP-C at Ser282, and SS-31 treatment normalized this age-related decrease in cMyBP-C Ser282 phosphorylation (*Figure 6a*), consistent with its association with improved relaxation. Cardiac troponin I (cTnI) is an inhibitory subunit of troponin, and phosphorylation of cTnI has been shown to increase the rate of cardiac relaxation (*Zhang et al., 1995*). Phosphorylation of Ser23/24 and Ser150 of cTnI was not altered in old murine hearts, and SS-31 treatment had no effect on Ser23/24 and Ser150 phosphorylation (*Figure 6b and c*). Titin is a giant myofilament protein in the sarcomere and titin isoform ratio (N2BA/N2B ratio) can modulate passive myocardial and diastolic function (*Nagueh et al., 2004*). However, we observed no changes in N2BA/N2B ratio with SS-31 treatment (*Figure 6d*).

## Late-life mCAT expression also improved diastolic function and SS-31 treatment cannot further improve cardiac function in old mCAT mice

To determine if reducing mtROS in late-life is sufficient to rescue age-related cardiac dysfunction, we administered an adeno-associated virus serotype-9 vector expressing mitochondrial-targeted catalase (AAV9-mCAT) (*Li et al., 2009*) to old C57Bl/6 mice to induce expression of catalase in cardiac mitochondria. We observed improved diastolic function at 12 weeks after AAV9-mCAT administration (*Figure 7a*), suggesting that late-life reduction of mtROS is sufficient to initiate the molecular changes required to reverse age-related diastolic dysfunction. Life-long expression of mCAT was previously shown to prevent age-related mitochondrial ROS accumulation and substantially attenuate declines in cardiac function in old mCAT mice (*Dai et al., 2009*). To determine if SS-31 treatment would have additive impact on mCAT mice, we administered SS-31 to old mCAT mice, but observed no further improvement in diastolic function at up to 8 weeks (*Figure 7b*), although the SS-31 induced improvement in diastolic function seen previously in old wild-type mice was fully recapitulated. These results suggest that the cardiac benefits induced by SS-31 and mCAT are mechanistically overlapping.

## SS-31 treatment and mCAT expression have differential effects on myofilament protein phosphorylation

To investigate the mechanism by which mCAT expression improves diastolic function, we assessed how late-life mCAT expression altered phosphorylation of myofilament proteins. Unlike SS-31, late-life mCAT expression resulted in slight reduction in Ser282 phosphorylation of cMyBP-C (*Figure 7c*). Interestingly, late-life mCAT expression increased phosphorylation of cTnI at Ser23/24 and Ser150

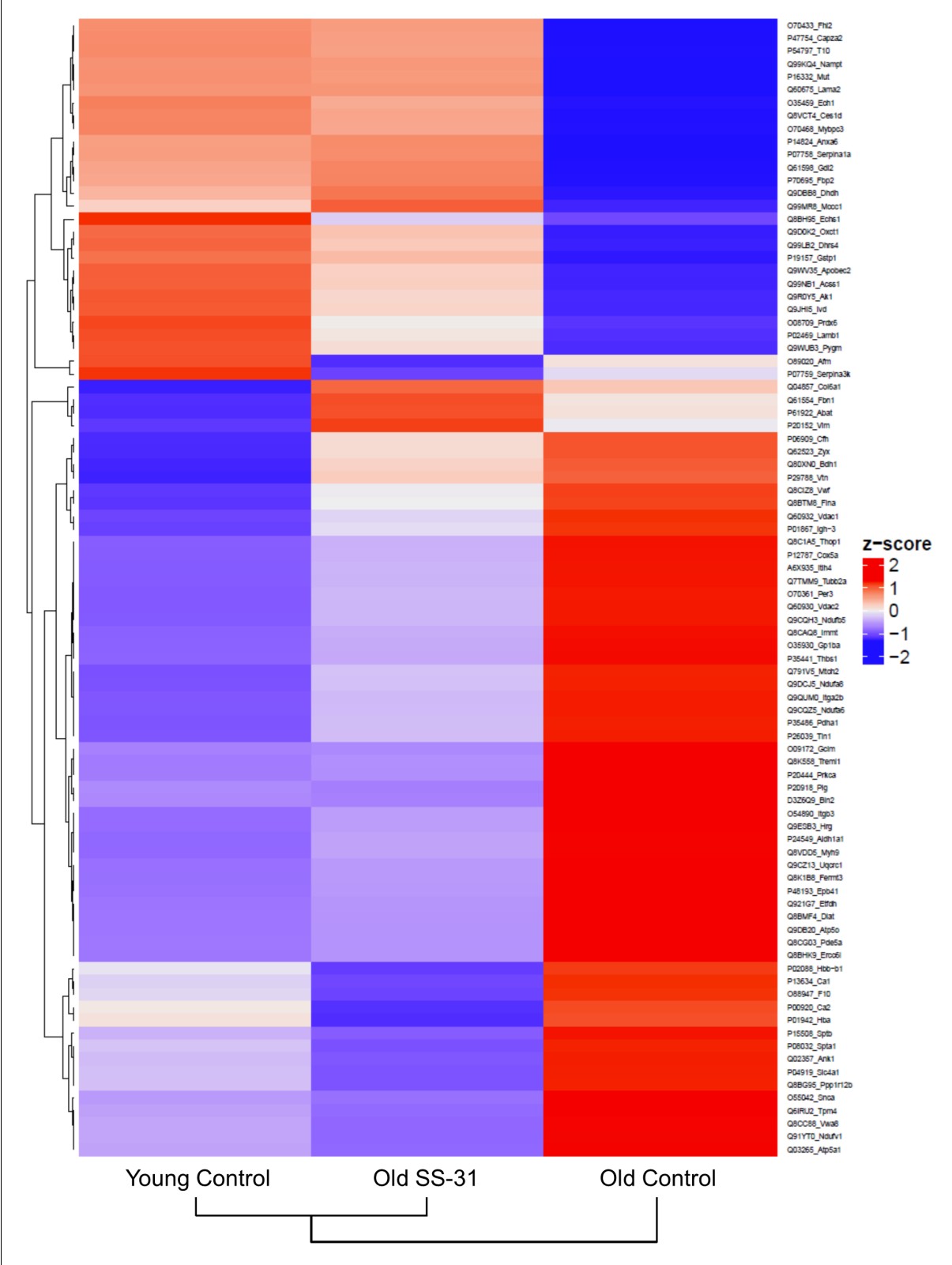

**Figure 5.** SS-31 treatment partially restores age-related proteomic remodeling. A heatmap of z-scores the 88 proteins that were significantly altered by both aging (q < 0.05 for old control vs. young control) and SS-31 treatment (q < 0.05 for old SS-31 vs. old control); n = 9, 10, and eight male mice for young control, old control and old SS-31, respectively, analyzed as described in the method section. We computed the z-scores of the average log2 abundance values for each of the three groups, where we adjusted the data, by protein, to have a mean of zero and a standard deviation of 1. The

*Figure 5 continued on next page*

*Figure 5 continued*

heatmap was generated using the ComplexHeatmap (v.1.20.0) R package (*Gu et al., 2016*), where both the sample groups and the proteins were clustered via the hclust function with the 'complete' agglomeration method. Distance matrix for clustering were computed using 'Euclidean' distance. The resulting heatmap presents the proteins in rows and sample groups in columns, both of which were grouped according to the clustering results. Row labels on the right are the UniProt ID_Gene Name of each protein. The identities and fold changes of all protein identified are listed in **Supplementary file 3**.

The online version of this article includes the following figure supplement(s) for figure 5:

**Figure supplement 1.** SS-31 treatment induces modest changes in metabolome that partially attenuates the age-related changes.

**Figure supplement 2.** A network of metabolite set enrichment for the 11 metabolites that were significantly (p<0.05, by Tukey's HSD) altered by aging.

(*Figure 7d and e*), which may contribute to the improved diastolic function. While SS-31 treatment and mCAT expression have differential effects on regulation of myofilament protein phosphorylation, both interventions mediate improved diastolic function in old mice.

## Discussion

Mitochondrial dysfunction is a hallmark of aging and has been implicated in the pathogenesis of cardiovascular diseases. We tested the hypothesis that pharmacologic targeting of mitochondrial dysfunction in late-life can reverse age-related cardiac dysfunction in mice. The main findings of the study are: 1) enhancing mitochondrial function at late-life by administration of mitochondrial-targeted SS-31 peptide or AAV-mediated expression of mitochondrial targeted catalase can reverse pre-existing cardiac dysfunction in old mice; 2) SS-31 treatment normalizes the age-related increase in mitochondrial proton leak, reduces ROS production by old cardiomyocytes, and reduces protein oxidative modifications; 3) the rescue of diastolic function by SS-31 in old mice is due, at least in part, to reversal of hypo-phosphorylation of myofilament protein cMyBP-C; and 4) SS-31 treatment and mCAT expression, while similar in many ways, differentially regulate myofilament protein phosphorylation. These findings are summarized in a proposed mechanistic model of how SS-31 treatment and mCAT expression improve mitochondrial function and regulates myofilament properties to improve cardiomyocytes relaxation and reversing age-related cardiac dysfunction (*Figure 8*).

### Targeting mitochondrial oxidative stress in late-life reverses cardiac aging phenotypes

Transgenic mCAT expression reduces mitochondrial oxidative stress and attenuates cardiac aging phenotypes in mice (*Dai et al., 2009*). While life-long mCAT expression has many positive effects (*Dai et al., 2017*), including prevention of pressure-overload induced cardiac hypertrophy or failure (*Dai et al., 2011a*; *Dai et al., 2012*) and attenuating the decline in cardiac function during aging (*Dai et al., 2009*), there may be negative pleotropic effects at young age (*Basisty et al., 2016*). For this, and practical reasons, a treatment that can be started at old age to reverse cardiac aging is a much more desirable therapeutic strategy. Here, we demonstrated that both SS-31 treatment and mCAT expression starting at late-life can reverse the age-related decline in diastolic function. This result suggests that reducing mitochondrial oxidative stress at late-life can be sufficient to initiate molecular changes, including phosphorylation of myofilament proteins, to improve diastolic function.

SS-31 (elamipretide) is a tetrapeptide with an alternating aromatic-cationic amino acids motif that is selectively enriched in mitochondria (*Szeto, 2006*). Although SS-31 was initially thought to be a mitochondrial targeted antioxidant (*Cho et al., 2007a*; *Cho et al., 2007b*), it has more recently been shown to interact with cardiolipin to enhance inner membrane cristae curvature and function of the electron transport chain (ETC), including the electron carrying activity of cyt c, while reducing cyt c peroxidase activity (*Birk et al., 2014*; *Birk et al., 2013*; *Szeto, 2013*). A single injection of SS-31 to old mice has been shown to enhance mitochondrial energetics in skeletal muscle (*Siegel et al., 2013*). Subcutaneous delivery of SS-31 for 8 weeks improves redox homeostasis and mitochondrial function in skeletal muscle and enhances exercise tolerance (*Campbell et al., 2019*), and the same regimen also improves kidney glomerular architecture in old mice (*Sweetwyne et al., 2017*). 2-week daily injection of SS-31 restores neurovascular coupling responses and improves cognition in old mice (*Tarantini et al., 2018*). In addition to these positive effects in normal aging, SS-31 treatment has been shown to protect against cardiovascular diseases. SS-31 treatment offers

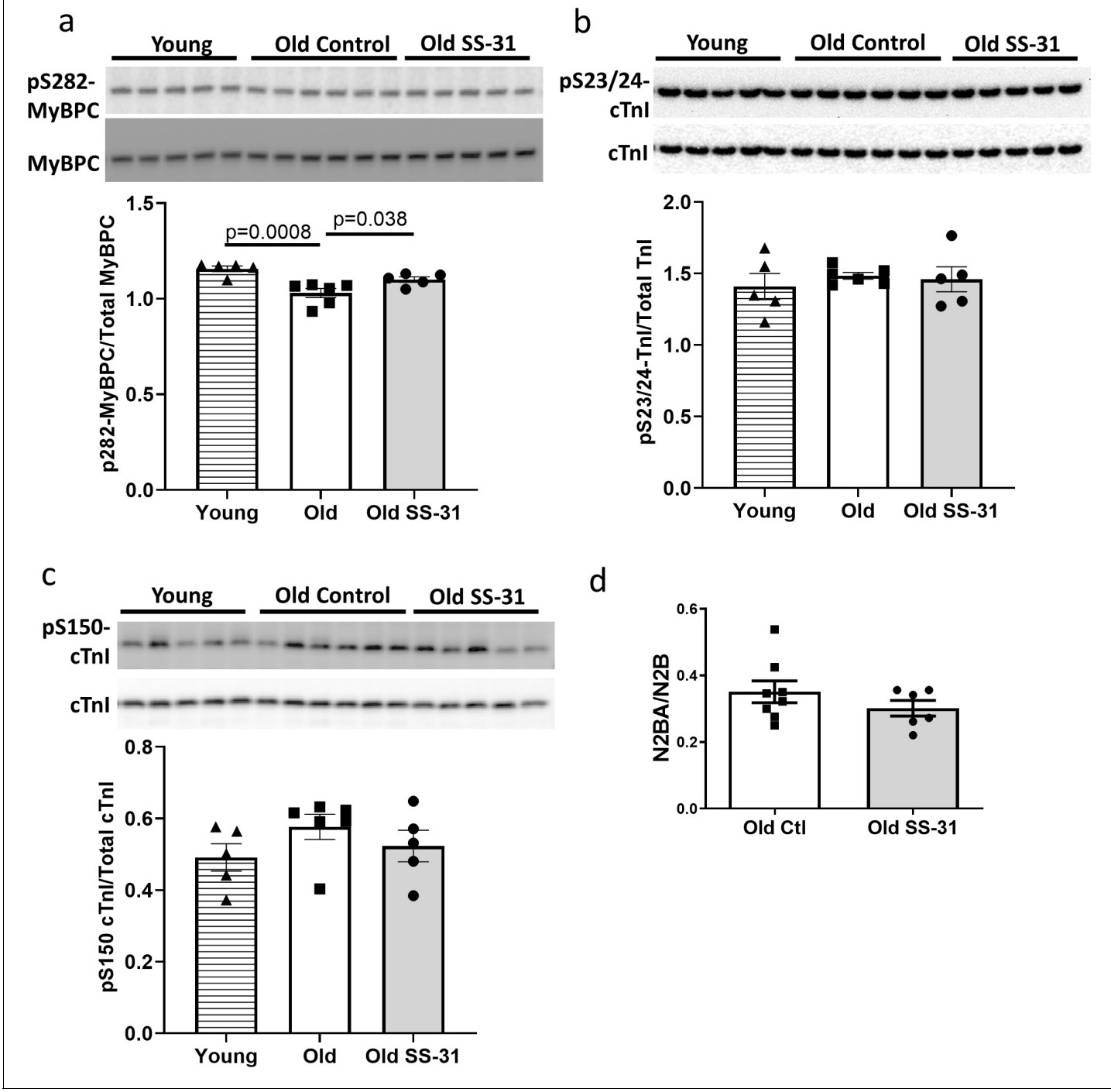

**Figure 6.** SS-31 rescues the age-related hypo-phosphorylation of MyBP-C. (**a**) Old murine hearts displayed reduced levels of MyBP-C phosphorylation at Ser282, which is normalized by SS-31 treatment. (**b–c**) Aging and SS-31 treatment did not alter phosphorylation of cTnI at Ser23/24 (**b**) and Ser150 (**c**) in hearts For panel a-c, n = 5, 6, and five male mice for young control, old control and old SS-31, respectively, analyzed by one-way ANOVA Dunnett's post hoc analysis for panel a-c. (**d**) Titin isoform ratio (N2BA/N2B ratio) did not change with SS-31 treatment; n = 8 and 6 female mice were used for old control and old SS-31, respectively, and were compared by unpaired T-test.

cardioprotection in models of cardiac ischemia-reperfusion and myocardial infarction (*Cho et al., 2007a*; *Brown et al., 2014*; *Dai et al., 2014c*; *Kloner et al., 2012*; *Shi et al., 2015*; *Szeto, 2008*), it also prevents cardiac fibrosis and hypertrophy induced by 4-week Angiotensin II infusion (*Dai et al., 2011b*). As with mCAT expression, SS-31 ameliorates cardiac fibrosis and improves cardiac function

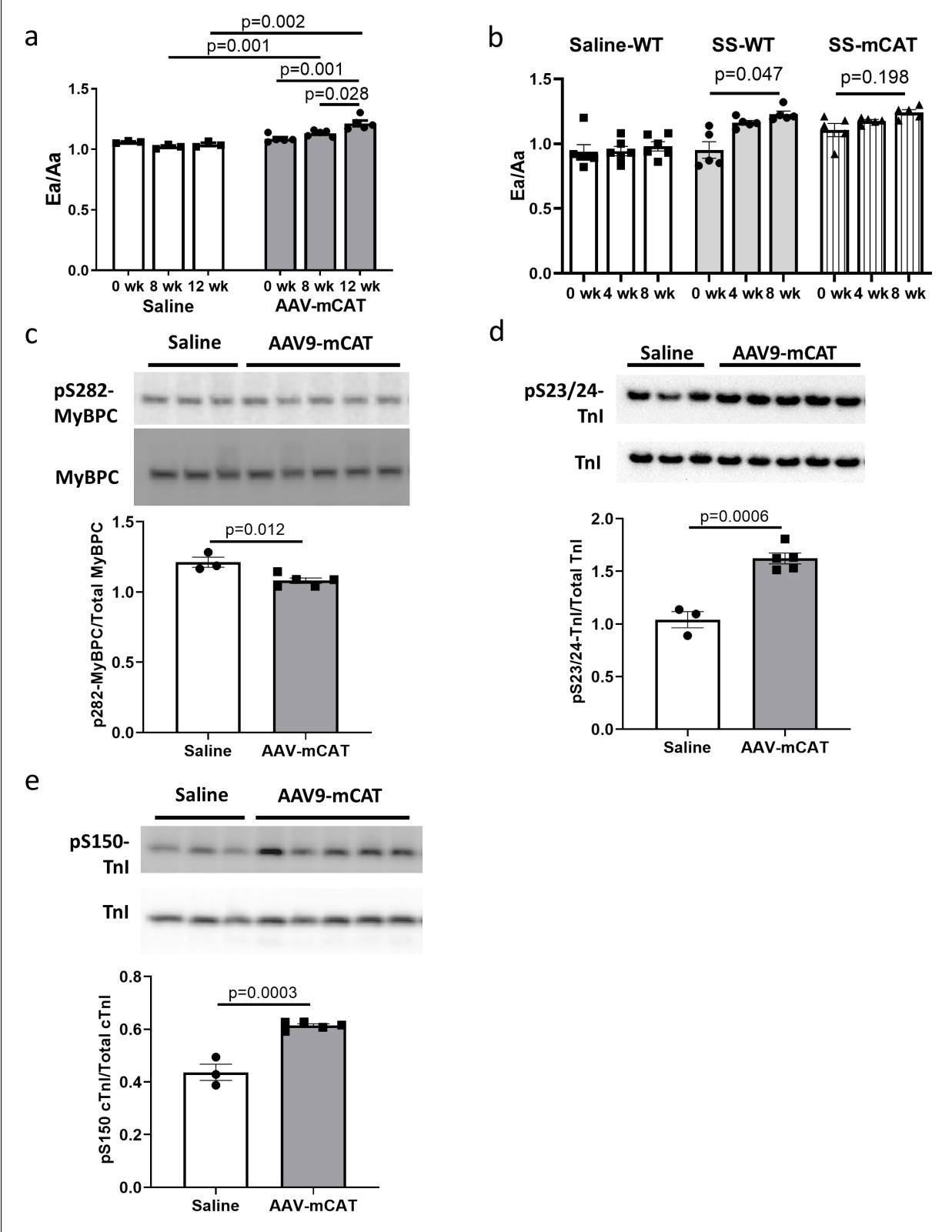

**Figure 7.** The cardiac benefit of SS-31 treatment is not additive to that of mCAT expression but the two interventions differentially regulate myofilament protein phosphorylation. (a) Diastolic function (Ea/Aa) improved at both 8 and 12 weeks after AAV9-mCAT administration. n = 3 (saline) and n = 5 (AAV-mCAT) female mice were analyzed by repeated measure ANOVA with Tukey's multiple comparison test between time points and Sidak post hoc analysis between treatment groups. (b) 8-week SS-31 improved diastolic function in old WT but did not further improve the function of old

*Figure 7 continued on next page*

*Figure 7 continued*

mCAT mice; n = 5 (for SS-WT and SS-mCAT) and n = 6 (for Saline-WT) mixed-sex mice were analyzed by repeated measure ANOVA with Tukey's multiple comparison test between time points. (c) Late-life mCAT expression reduced Ser282 phosphorylation of MyBP-C. (d–e) Late-life mCAT expression increased phosphorylation of cTnI at Ser23/24 (d) and Ser150 (e). For panel c-e, n = 3 and 5 female mice were used for saline and AAV-mCAT, respectively, and were analyzed by unpaired T-test.

in pressure overload-induced heart failure, and proteomic analyses revealed that both SS-31 and mCAT expression attenuated changes in proteins related to mitochondrial function (*Dai et al., 2013*; *Dai et al., 2012*). However, whether SS-31 treatment is protective against cardiac aging had not been previously established.

In this study we found that the reversal of cardiac aging phenotypes by late-life SS-31 treatment was accompanied by reduced oxidative protein modifications in aged hearts, which can be explained by the reduced mitochondrial superoxide and hydrogen peroxide production in SS-31 treated cardiomyocytes. As shown in *Figure 8*, SS-31 and mCAT both reduce ROS, however, the former is believed to do so by prevention of ROS production by electron transport chain, while the latter directly scavenges hydrogen peroxide. Both, however, will inhibit the vicious cycle of ROS induced damage to mitochondrial DNA and proteins and prevent pathological ROS-Induced-Redox Signaling (*Figure 8*). Consistent with this overlap, SS-31 treatment cannot further improve the cardiac function of old mCAT mice (*Figure 7b*), supporting the role of reduction of mitochondrial oxidative stress as a key mechanism of SS-31 reversal of cardiac aging. Although 8-week SS-31 treatment did not significantly further improve diastolic function in old mCAT mice, there appeared to be an increasing trend and it is possible that longer duration of treatment might lead to further additive improvement. This could be explained by the involvement of other molecular mechanisms, especially those that may be due to direct augmentation of mitochondrial ATP production by SS-31 (*Figure 8*).

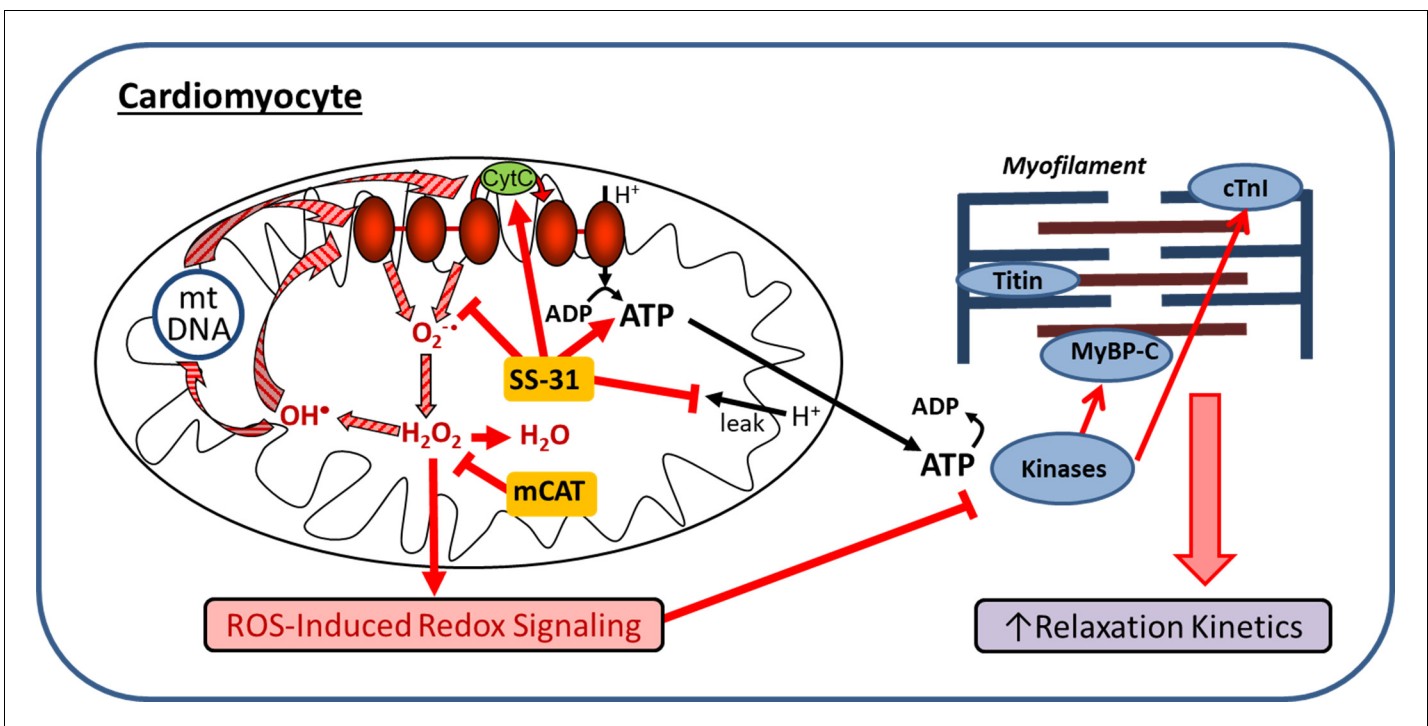

**Figure 8.** Schematic outline of results and interpretation. While mCAT and SS-31 both inhibit electron transport chain produced ROS, they do so by different mechanisms. Both inhibit a ROS-mediated vicious cycle (ROS induced mtDNA and protein damage leads to greater ROS generation; striped arrows) and ROS-Induced redox signaling. However, by promoting electron transport, preventing proton leakage and augmenting ATP production, SS-31 also improves mitochondrial energetics. By improving mitochondrial energetics and reducing pathologic redox signaling, SS-31 promotes phosphorylation of cMyBP-C to enhance myofilament relaxation kinetics, while mCAT expression does so through promoting phosphorylation of cTnI.

In contrast to the benefits detected in old mice after SS-31 treatment, we did not observe significant differences in the parameters we measured in young mice with SS-31 treatment, including exercise performance (*Figure 1e*), cardiac hypertrophy (*Figure 1d*) and cardiac metabolome (*Figure 5— figure supplement 1*). These results suggest that SS-31 treatment is effective in aged hearts with pre-existing mitochondrial dysfunction but has little effects in young hearts with normal functioning mitochondria. The absence of SS-31 induced improvement in healthy mitochondria has previously been reported by studies of diseases models and aging skeletal muscle (*Campbell et al., 2019*; *Siegel et al., 2013*; *Szeto, 2013*), potentially due to the inability to further increase ATP production in individuals with optimal mitochondrial function.

A recent study by Cieslik and colleagues showed that combined treatment with glutathione precursors, N-Acetyl Cysteine (NAC) and Glycine, but not NAC alone, can reverse age-related diastolic dysfunction (*Cieslik et al., 2018*). While the study suggests that aged mice may benefit from increased glutathione content, the effects of the combined treatment on in vivo cardiac glutathione contents and S-glutathionylation remain to be established. Here, we showed that SS-31 can reverse diastolic dysfunction and normalized the age-related increase in protein S-glutathionylation without the necessity to exogenously alter glutathione levels. This supports the primary importance of reducing mitochondrial ROS in redox homeostasis. A follow-up protein-by-protein analysis of SS-31 effects on S-glutathionylation will provide further insights on how SS-31 regulates redox homeostasis in the aging hearts (unpublished). In addition, the fact that SS-31 treatment or late-life mCAT expression can reverse age-related diastolic dysfunction but NAC alone fails to do so supports the importance of mitochondrial localization of the ROS combating action in cardiac aging protection.

## Normalization of proton leak in aged cardiomyocytes is a protective mechanism of SS-31

Mitochondrial oxidative phosphorylation is the major source of ATP production in the cell. When electrons from substrate oxidation pass through electron transport chain complexes, the energy generated is used to pump protons from the mitochondrial matrix to the intermembrane space to generate a proton gradient. The resulting protonmotive force drives protons back to mitochondrial matrix through ATP synthase, while converting ADP to ATP. However, this coupling of substrate oxidation and ATP synthesis is incomplete as protons can also re-enter mitochondrial matrix independent of ATP synthesis in a process termed 'proton leak' (*Jastroch et al., 2010*). In this study, we showed that cardiomyocytes from old mice exhibited increased proton leak when compared to cardiomyocytes from young mice. This age-related increase in proton leak is consistent with previous observations that aging increases proton leak in mouse hepatocytes and in mitochondria from rat heart, kidney and liver (*Harper et al., 1998*; *Serviddio et al., 2007*). The increased proton leak implicates that aging reduces coupling of substrate oxidation to ATP synthesis, and agrees with the reduced mitochondrial coupling observed in aged skeletal muscle (*Campbell et al., 2019*; *Siegel et al., 2013*). Strikingly, 8-week treatment with SS-31 completely reversed the age-related increase in mitochondrial proton leak in cardiomyocytes (*Figures 2c, e* and *8*).

While the molecular mechanisms of proton leak are not fully understood, it has been shown that basal leak through mitochondrial inner membrane or around inner membrane proteins, and inducible leak through adenine nucleotide translocase (ANT) or uncoupling proteins (UCPs) contribute to mitochondrial proton leak (*Jastroch et al., 2010*). Because SS-31 interacts with cardiolipin in the inner mitochondrial membrane, it is possible that SS-31 can regulate basal leak by preserving inner membrane integrity and normalizing the function of inner membrane spanning proteins. ROS has been shown to induce mitochondrial uncoupling and increase proton leak (*Brookes and Mitochondrial, 2005*; *Brookes et al., 1998*; *Echtay et al., 2002*), and thus, reduced ROS production following SS-31 treatment may also contribute to the lowered proton leak. A protective function of ROS-induced proton leak has been suggested, where increased ROS level promotes proton leak to reduce mitochondrial membrane potential and decrease further ROS production and oxidative damage (*Brookes and Mitochondrial, 2005*). However, the reduced RCR and increased oxidative damage seen in the aged heart suggest that this increased proton leak is maladaptive and a result of compromised mitochondria function. In addition to suppressing proton leak, SS-31 partially restores the RCR and increases mitochondrial membrane potential in aged cardiomyocytes, suggesting that SS-31 treatment reverses age-related mitochondrial dysfunction. It has been shown that mitochondrial dysfunction induces cellular senescence, and PolG mutator mice, which have increased mtDNA

mutation and mtROS, also have accumulation of senescent cells (*Wiley et al., 2016*). Compared to old controls, SS-31 treated hearts have reduced numbers of p16- or p19-positive senescent cells and this may be a direct result of the restoration of mitochondrial function (*Wiley et al., 2016*).

## SS-31 and mCAT expression differentially regulate phosphorylation of myofilament proteins to improve diastolic function

SS-31 treatment reduces cardiac dysfunction in models of pressure-overload induced heart failure, and this is accompanied by marked proteomic changes (*Dai et al., 2013*). In comparison, however, SS-31 induces more modest changes in protein expression in the aging heart (*Figure 5*), and no changes in expression of OXPHOS subunits (*Figure 3*). In this study, we also investigated the effect of SS-31 treatment on the cardiac metabolite profile and detected modest changes in metabolite levels, with SS-31 treatment showing a tendency of attenuation of the age-related metabolic changes.

On the other hand, post-translational modifications, both oxidative and phosphorylative, may play a more important role in conferring the benefits of SS-31 treatment. In cardiac muscle, the state of post-translational modifications of myofilament proteins are crucial to the regulation of contractile and relaxation behaviors, as shown in the pathophysiology of heart failure (*Biesiadecki, 2016*; *Hamdani et al., 2013a*; *Ramirez-Correa et al., 2014*). In particular, phosphorylation of myofilament proteins is a key modulator of diastolic function of the heart (*Hamdani et al., 2013a*; *Hamdani et al., 2013b*; *Rosas et al., 2015*). cMyBP-C is a sarcomeric protein that modulates actin–myosin interaction and cross-bridge cycling. It is a critical mediator of diastolic function whose activity has been shown to be regulated by phosphorylation of its cardiac specific M-domain (*Tong et al., 2014*; *Rosas et al., 2015*). Previous studies have shown that a high level of phosphorylation is critical to normal cardiac function, and hypo-phosphorylation of the M-domain is associated with heart failure (*Copeland et al., 2010*; *Jacques et al., 2008*; *Kooij et al., 2013*). Ser282 is one of the phosphorylation sites in the M-domain. Sadayappan and colleagues showed that Ser282 phosphorylation is critical for subsequent phosphorylation of Ser302, another phosphorylation site in the M-domain of cMyBP-C, and phospho-ablation at Ser282 impairs baseline diastolic function and response to β-adrenergic stimulation (*Sadayappan et al., 2011*). We detected an age-related decrease in phosphorylation of S282 in the M-domain of cMyBP-C, and SS-31 treatment restores Ser282 phosphorylation in old mice, likely contributing to the restoration of diastolic function (*Figures 6a* and *8*).

TnI is an inhibitory subunit of troponin, and it binds to actin to inhibit actomyosin interaction in the absence of calcium binding to TnC (*Gomes et al., 2002*). Phosphorylation of cTnI at Ser23/24 by protein kinase A has been shown to reduce myofilament $Ca^{2+}$ sensitivity and increase myofilament relaxation rate (*Zhang et al., 1995*; *Kentish et al., 2001*). Biesiadecki lab recently showed simultaneous increases in Ser23/24 and Ser150 phosphorylation in ischemic hearts and suggested that this combined phosphorylation plays an adaptive role in ischemia by maintaining $Ca^{2+}$ sensitivity and accelerating $Ca^{2+}$ dissociation (*Nixon et al., 2014*; *Salhi et al., 2016*). Late-life mCAT expression, but not SS-31, increased phosphorylation of cTnI at Ser23/24 and Ser150 (*Figures 7d, e* and *8*). The effects of these increases in cTnI phosphorylation on $Ca^{2+}$ sensitivity and myofilament relaxation in aging cardiac muscles remain to be investigated. Unlike SS-31 treatment, late-life mCAT expression fails to increase Ser282 phosphorylation of cMyBP-C. Although reduction of mtROS is a shared protective mechanism between SS-31 and mCAT, it is likely that the two interventions activate different kinases to regulate myofilament protein phosphorylation to mediate improved relaxation in the aging hearts. Future ex vivo biomechanical assays are required to determine how the two interventions differentially regulate cross-bridge kinetics and $Ca^{2+}$ sensitivity to improve diastolic function.

A limitation of this study is that due to the limited number of mice in the persistence cohort, we did not follow up on how different molecular changes responded after cessation of SS-31 treatment due to the limited number of mice in the persistence cohort. We note that the persistence of SS-31 induced functional benefit varied between individual mice (*Figure 1—figure supplement 3*), and thus, we hypothesize that the persistence of the molecular changes (e.g. oxidative modifications and myofilament protein phosphorylation) also varied between individuals. Future studies will be required to determine on how different molecular changes mediated by SS-31 persist after treatment cessation and identify the molecular mechanisms driving this individual variation and limiting SS-31 persistence.

In conclusion, this study demonstrated that late-life SS-31 treatment can reverse pre-existing cardiac aging phenotypes, including diastolic dysfunction. Besides reducing mitochondrial ROS production and oxidative damage, SS-31 treatment reduces the age-related increase in mitochondrial proton leak in cardiomyocytes. Despite similar cardiac benefits, SS-31 and mCAT expression induced differential changes in myofilament protein phosphorylation, in line with overlapping but not concordant mechanisms of action. These results support the therapeutic potential of targeting mitochondrial dysfunction to reverse the effects of cardiac aging.

# Materials and methods

## Key resources table

| Reagent type (species) or resource | Designation | Source or reference | Identifiers | Additional information |
|---|---|---|---|---|
| Strain, strain background (*M. musculus*; male and female) | C57BL/6J | National Institute of Aging Charles River colony | RRID:IMSR_JAX:000664 | |
| Genetic reagent (*M. musculus*) | mCAT | Rabinovitch Lab; PMID:15879174 | RRID:IMSR_JAX:016197; *Tg(CAG-OTC/CAT)4033Prab* | Now available at The Jackson Lab |
| Transfected construct (*M. musculus*) | AAV9-mCAT | Duan Lab, PMID:19690612 | AV.RSV.MCAT | Adeno-associated virus construct to transfect and express mCAT transgene |
| Antibody | Anti-OXPHOS (Rabbit polyclonal) | Abcam | ab110413 RRID:AB_2629281 | (1:500) |
| Antibody | Anti-Troponin I (Rabbit polyclonal) | Cell Signaling Technology | #4002 RRID:AB_2206278 | (1:1000) |
| Antibody | Anti-pSer23/24-Troponin I (Rabbit polyclonal) | Cell Signaling Technology | #4004 RRID:AB_2206275 | (1:1000) |
| Antibody | Anti-pSer150-Troponin I (Rabbit polyclonal) | ThermoFisher | PA5-35410 RRID:AB_2552720 | (1:1000) |
| Antibody | Anti- cMyBP-C (Mouse monoclonal) | Santa Cruz SC-137237 | SC-137237 RRID:AB_2148327 | (1:1000) |
| Antibody | Anti-pSer282-cMyBP-C (Rabbit polyclonal) | Enzo | ALX-215–057 R050 RRID:AB_2050502 | (1:2000) |
| Antibody | anti-p19 (Rabbit polyclonal) | LSBio | LS-C49180 RRID:AB_1192824 | (1:300) |
| Antibody | anti-p16 (Rabbit polyclonal) | Abcam | ab211542 | (1:300) |
| Antibody | Donkey anti-Rabbit Secondary Antibody, HRP | ThermoFisher | A16035 RRID:AB_2534709 | (1:10000) |
| Antibody | Goat anti-Mouse Secondary Antibody, HRP | ThermoFisher | A16072 RRID:AB_2534745 | (1:10000) |
| Peptide, recombinant protein | SS-31 peptide (Elamipretide) | Stealth Bio Therapeutics | | 3 µg/g body weight/day |
| Commercial assay or kit | OxiSelect protein carbonyl ELISA kit | Cell Biolabs | STA-310 | |
| Commercial assay or kit | ImmPRESS-VR Anti-Rabbit IgG HRP Polymer Detection Kit | Vector Laboratories | MP-6401–15 | |
| Commercial assay or kit | Seahorse XF Cell Mito Stress Test Kit | Aligent/Seahorse Bioscience | 103015–100 | |

*Continued on next page*

*Continued*

| Reagent type (species) or resource | Designation | Source or reference | Identifiers | Additional information |
|---|---|---|---|---|
| Commercial assay or kit | MitoSOX Red | ThermoFisher | M36008 | |
| Commercial assay or kit | MitoPY1 | Fisher Scientific/Tocris Bioscience | 44-281-0 | |
| Commercial assay or kit | MitoTracker Green | ThermoFisher | M7514 | |
| Commercial assay or kit | MitoTracker Deep Red | ThermoFisher | M22426 | |
| Commercial assay or kit | JC-1 Dye | ThermoFisher | T3168 | |
| Commercial assay or kit | BCA protein assay | Thermo Scientific | 23225 | |
| Commercial assay or kit | Pierce Reversible Protein Stain Kit for PVDF Membranes | Thermo Scientific | 24585 | |
| Commercial assay or kit | SuperSignal West Pico PLUS Chemiluminescent Substrate | Thermo Scientific | 34580 | |
| Software, algorithm | Graphpad Prism | Graphpad | RRID:SCR_002798 | |
| Software, algorithm | AlphaView Software | ProteinSimple | | |
| Software, algorithm | Metaboanalyst 4.0 | www.metaboanalyst.ca; PMID:29762782 | RRID:SCR_015539 | |
| Software, algorithm | Topograph | MacCoss Lab software; PMID:22865922 | | |
| Software, algorithm | ComplexHeatmap | https://github.com/jokergoo /ComplexHeatmap PMID:27207943 | RRID:SCR_017270 | |

## Animals

Young (3–5 month-old) and old (24-month-old) C57BL/6 male and female mice were obtained from the National Institute of Aging Charles River colony. All mice were handled according to the guidelines of the Institutional Animal Care and Use Committee of the University of Washington. Mice were housed at 20°C in an AAALAC accredited facility under Institutional Animal Care Committee supervision.

For each sex, old mice were randomly assigned to two groups and SS-31 (3 μg/g body weight/day; kindly provided by Stealth BioTherapeutics, Newton, MA) or saline-vehicle were delivered subcutaneously via osmotic minipumps (Alzet 1004) for 4 weeks. After 4 week, the original minipump was surgically removed and a new minipump was implanted to continue the SS-31 or saline-vehicle delivery for another 4 weeks. For evaluation of persistent effects of SS-31 treatment, the minipump was surgically removed after 8-week treatment.

To study the effect of reducing mtROS at late-life, recombinant AAV9 vector expressing a mitochondria-targeted catalase gene (AAV9-mCAT) was delivered to 24-month-old WT C57BL/6 female mice by retro-orbital injection. A total of $5 \times 10^{12}$ vg particles of AAV were delivered to each mouse (*Li et al., 2009*). Retro-orbital injection of saline was performed as control.

To study the interaction of SS-31 and catalase, 23- to 27-month-old mixed-sex mCAT mice and age-matched WT littermates were given 8-week subcutaneous delivery of SS-31 (3 μg/g body weight/day) or saline-vehicle via minipumps.

Echocardiography was performed at baseline and post-treatment timepoints to evaluate systolic and diastolic function of the mice. Mouse was anesthetized by 0.5–1% isoflurane and echocardiogram was performed using a Siemens Acuson CV-70 equipped with a 13MHz probe.

For treadmill running, male mice were acclimated to the treadmill for two consecutive days before the measurement. At the day of measurement, mice were placed on the treadmill at a 10° incline when the treadmill accelerated from 0 m/min to 30 m/min in a 5 min period, and allowed to run to exhaustion at 30 m/min. Exhaustion was determined if the mice fail to remount the treadmill after receiving five consecutive shocks and light physical prodding. Treadmill measurements were performed during the natural active period of the mice between 8 pm and two am.

At the endpoint, mice were euthanized by cervical dislocation. The heart was immediately removed, weighed and processed for downstream analyses.

## Cardiomyocyte isolation from adult mice

Ventricular myocytes were enzymatically isolated from the hearts of C57BL/6 mice using a protocol modified from that described previously (*Zhang et al., 2013*). Briefly, the animal was euthanized by cervical dislocation. The heart was immediately removed from the chest, raised and perfused with oxygenated modified Ca2+ free-Tyrode's solution for 5 min. Then the heart was perfused with 50 ml low Ca2+ solution containing 300 U/ml collagenase II + 0.5 mg/ml hyaluronidase at 37°C for 20–30 min. The ventricle was cut into small pieces and further digested under gentle agitation. Rod-shaped adult cardiomyocytes were collected by settling down of cells and plated in 24 well plates for XF24e Extracellular Flux Analyzer analysis (Seahorse Bioscience) or on glass coverslips for confocal imaging.

## Cardiomyocyte imaging

For confocal imaging, we used modified Tyrode's solution (in mM: 138 NaCl, 0.5 KCl, 20 HEPES, 1.2 MgSO4, 1.2 KH2PO4, 1 CaCl2, 5 Glucose, pH 7.4) and a Leica SP8 (Leica, Germany) inverted confocal microscope for confocal imaging at room temperature. For mitochondrial superoxide quantitation, we used the ratio of MitoSOX Red (5 µM, excited at 540 nm and emissions collected at >560 nm) to MitoTracker Green (200 nM, excited at 488 nm and emission collected at 505–530 nm). For mitochondrial H2O2 measurement, we used the ratio of MitoPY1 (5 µM, excited at 488 nm and emission collected at 520–640 nm) and MitoTracker Deep Red (100 nM, excited at 633 nm and emission collected >660 nm). For mitochondrial membrane potential measurement, JC-1 was excited by 488 nm laser and emission collected at 510–545 nm and 570–650 nm.

## Cardiomyocyte respiration measurement

For intact cardiomyocyte respiration measurement, 800 cardiomyocytes were plated in each well of XF24e Extracellular Flux Analyzer 24 well plates (Seahorse Bioscience) and mitochondrial respiration was assessed in 3 hr after plating. Mitochondrial respiration was assessed using the Seahorse Bioscience XF Cell Mito Stress Test assay, with OCR values measured at baseline and after the sequential addition of 1 µM oligomycin, 0.5 mM FCCP and 1 µM rotenone +1 µM antimycin A (*Zhang et al., 2017*). The OCR values for basal respiration, proton leak, ATP turnover, maximum respiration and non-mitochondrial respiration were thereby determined. Respiratory control ratio (RCR) was calculated as the ratio of maximum respiration to proton leak.

## Immunoblotting

Proteins were extracted from frozen heart tissues with K150T buffer (150 mM KCl, 50 mM Tris-HCl pH7.4, 0.125% Na deoxycholate, 0.375%Triton X-100, 0.15% NP-40, 4 mM EDTA, 50 mM NaF) and quantified by BCA protein assay (Thermo Scientific). Equal amount of proteins (15 µg) were resolved on 4–12% NuPAGE Bis-Tris gel and transferred to PVDF membrane. A Pierce Reversible Protein Stain Kit was used to detect total proteins for normalization of loading.

Primary antibodies used in immunoblotting were OXPHOS (Abcam ab110413, at 1:500), Troponin I (Cell Signaling Technology #4002, 1:1000), pSer23/24-Troponin I (Cell Signaling Technology #4004, 1:1000), pSer150-Troponin I (ThermoFisher PA5-35410; 1:1000), cMyBP-C (Santa Cruz SC-137237, 1:1000), pSer282-cMyBP-C (ALX-215–057 R050, 1:2000).

Secondary antibodies used were donkey anti-rabbit IgG secondary antibody and goat anti-mouse IgG secondary antibody (both from Thermo Scientific). SuperSignal West Pico Chemiluminescent Substrate was used for detection and AlphaView Software (Protein Simple, San Jose, CA), was used for image acquisition and quantification.

## Measurement of protein S-glutathionylation

Quantification of protein-S-glutathionylation was performed using an established redox proteomics workflow (*Kramer et al., 2018*). Briefly, nine heart samples (young, aged with SS-31 treatment, and aged control, n = 3 for each) were subjected to protein extraction, selective reduction and enrichment, trypsin digestion and isobaric labeling with tandem mass tags 10-plex. For occupancy analysis, the levels of total thiol were quantified in a pooled sample. Mass spectrometry was performed on a Q Exactive Plus (Thermo Fisher Scientific), and data processing was conducted as previously described (*Kramer et al., 2018*). The raw mass spectrometry files for proteomics analysis of S-glutathionylation were uploaded to MassIVE (massive.ucsd.edu) with an accession ID: MSV000085329.

## Protein carbonyl assay

Protein carbonyl levels in heart tissues were measured using OxiSelect protein carbonyl ELISA kit (Cell Biolabs, San Diego, CA) according to the manufacturer's instructions.

## Metabolite profiling measurement

Pulverized cardiac tissues were homogenized in 200 µl of water and 800 µl of methanol were added to the homogenates. The homogenates were incubated on dry ice for 30 min and then sonicated in ice water bath for 10 min. The homogenates were centrifuged at 13000 rpm at 4°C for 5 min and the soluble extracts were dried by speed vac. The extracts were reconstituted and analyzed by LC-MS as described (*Dai et al., 2014b*).

The results of metabolite profiling were analyzed using Metaboanalyst 4.0 (*Chong et al., 2018*). After normalizing to input tissue weight, the relative peak intensities of metabolites were median normalized, log transformed and auto-scaled. One-way ANOVA was used for comparisons among all groups and Tukey's HSD was used for pairwise comparisons. A heatmap was generated for all metabolites with significantly different levels among groups (FDR < 0.05). For the 11 metabolites that showed age-related changes in levels (p<0.05 by Tukey's HSD), enrichment analysis was performed by Metaboanalyst 4.0 using the Pathway-associated metabolite sets as library.

## Immunohistochemistry

Hearts were fixed overnight in 4% paraformaldehyde, paraffin embedded and 4 µm sections deparaffinized, treated with ethylenediamine tetraacetic acid (EDTA) buffer pH eight and incubated with rabbit anti-p16 antibody (1:300, Abcam ab211542) or anti-p19 antibody (1:300, LSBio LS-C49180), Seattle, WA) overnight at 4°C. Secondary antibody detection was performed with ImmPRESS VR Anti-Rabbit IgG HRP Polymer Detection Kit (Vector Laboratories, Burlingame, CA), developed with diaminobenzidine (Sigma-Aldrich, St. Louis, MO) and counterstained with Mayer's Hematoxylin (Sigma-Aldrich, St. Louis, MO). Positive nuclear stain was expressed as a percentage of p16 positive or p19 positive nuclei (brown) versus total nuclei (brown + blue).

## Mass spectrometry for proteomic analysis

Pulverized heart tissues were homogenized in ice-cold extraction buffer (250 mM sucrose, 1 mM EGTA, 10 mM HEPES, 10 mM Tris-HCl pH7.4). Lysates were centrifuged at 800 x g for 10 min to remove debris. Samples were trypsin digested and purified by MCX column (Waters). LC-MS/MS analysis was performed with a Waters nanoAcquity UPLC and a Thermo Scientific Q Exactive mass spectrometer. Topograph software was used for peptide abundance measurement as previously described (*Dai et al., 2014b*). The statistical analysis of relative protein abundance between experimental groups was performed using a linear model of peptide abundance to calculate fold changes of proteins between experimental groups using the R/Bioconductor software. The p-values were adjusted for multiple comparison with the Bioconductor package q-value (*Dai et al., 2014b*). The fold changes and statistics of all identified proteins were shown in *Supplementary file 3*. In order to generate the heatmap, we computed a z-score of the average log2-abundance, where we adjusted the data, by protein, to have a mean of zero and a standard deviation of 1. The heatmap was generated using the Complex Heatmap (v.1.20.0) R package. We used IPA (https://www.qiagenbioinformatics.com/products/ingenuity-pathway-analysis/) to identify pathways that were significantly altered by both aging and SS-31 treatment within the dataset. The raw mass spectrometry files were uploaded to MassIVE (massive.ucsd.edu/) with an accession ID: MSV000084961.

## Measurement of titin isoforms

Relative expression of N2B and N2BA isoforms of titin was assessed in heart tissues using a vertical SDS-agarose gel system as previously described (*Tonino et al., 2017*; *Warren et al., 2003*). The ratio of intensities of N2BA band and N2B band was then determined.

## Statistical analyses

Echocardiographic results were analyzed by repeated measure ANOVA with Tukey's multiple comparison test between time points and Sidak post hoc analysis between treatment groups. Results of cardiomyocyte imaging, immunohistochemistry and AAV9-mCAT immunoblotting were analyzed by unpaired T-test compared to old saline control. HW/Tibia, mitochondrial respiration and immunoblotting results of SS-31 experiments were analyzed by one-way ANOVA with SNK or Dunnett's post hoc analysis. Graphpad Prism 8 was used for statistical analyses and data were plotted as mean with SEM. Results of metabolite profiling and proteomic analysis were analyzed as described in above sections. Data from mice that died before the designed endpoints were excluded from the study.

## Acknowledgements

The authors wish to acknowledge Stealth BioTherapeutics (Newton, MA) for kindly providing the SS-31 peptide. We thank Jeanne Fredrickson her assistance in biochemical assay and the Northwest Metabolomics Research Center (NW-MRC) for metabolomics service. We acknowledge funding support from Glenn/AFAR Postdoctoral Fellowship for Translational Research on Aging to YAC and HZ, NIA 5T32AG000057 Training Grant support and NIA K99/R00 AG051735 to YAC, AHA CDA 19CDA34660311 to HZ, NHLBI R35HL144998 (to HG), and NIA P01 AG001751 and P30 AG013280 to PSR. Redox proteomics experiments were performed in the Environmental Molecular Sciences Laboratory, Pacific Northwest National Laboratory, a national scientific user facility sponsored by the DOE under Contract DE-AC05-76RL0 1830.

## Additional information

### Funding

| Funder | Grant reference number | Author |
|---|---|---|
| Glenn Foundation for Medical Research | Glenn/AFAR Postdoctoral Fellowship Program for Translational Research on Aging | Ying Ann Chiao Huiliang Zhang |
| National Institute on Aging | 5T32AG000057 Training Grant | Ying Ann Chiao |
| National Institute on Aging | K99/R00 AG051735 | Ying Ann Chiao |
| National Institute on Aging | P01 AG001751 | Peter Rabinovitch |
| National Institute on Aging | P30 AG013280 | Peter Rabinovitch |
| American Heart Association | CDA 19CDA34660311 | Huiliang Zhang |
| National Heart, Lung, and Blood Institute | R35HL144998 | Henk L Granzier |

The funders had no role in study design, data collection and interpretation, or the decision to submit the work for publication.

### Author contributions

Ying Ann Chiao, Conceptualization, Data curation, Formal analysis, Funding acquisition, Investigation, Writing - original draft; Huiliang Zhang, Mariya Sweetwyne, Ellen Quarles, Matthew D Campbell, Tong Zhang, Gennifer Merrihew, Investigation, Writing - review and editing; Jeremy Whitson, Nathan Basisty, Lindsay K Pino, Lu Wang, Formal analysis, Writing - review and editing; Ying Sonia Ting, Formal analysis, Investigation; Ngoc-Han Nguyen, Investigation; Matthew J Gaffrey, Methodology; Yongping Yue, Dongsheng Duan, Resources; Henk L Granzier, Wei-Jun Qian, Methodology,

Writing - review and editing; Hazel H Szeto, Writing - review and editing; David Marcinek, Michael J MacCoss, Supervision, Writing - review and editing; Peter Rabinovitch, Conceptualization, Supervision, Funding acquisition, Writing - review and editing

### Author ORCIDs
Ying Ann Chiao  https://orcid.org/0000-0002-1256-4335
Henk L Granzier  http://orcid.org/0000-0002-9516-407X
Peter Rabinovitch  https://orcid.org/0000-0001-7169-3543

### Ethics
Animal experimentation: All mice were handled according to the guidelines of the Institutional Animal Care and Use Committee of the University of Washington and approved IACUC Protocol # 2174-23. Mice were housed at 20°C an AAALAC accredited facility under Institutional Animal Care Committee supervision.

### Decision letter and Author response
Decision letter https://doi.org/10.7554/eLife.55513.sa1
Author response https://doi.org/10.7554/eLife.55513.sa2

## Additional files

### Supplementary files
• Supplementary file 1. A dataset of relative abundances of all metabolites measured.

• Supplementary file 2. A table of all metabolites that showed significant differences in one or more comparisons.

• Supplementary file 3. A dataset of identities, fold changes and statistics of all proteins identified in the proteomic analysis.

• Transparent reporting form

### Data availability
Data file for metabolomic analysis (Table S1) and statistics of all proteins identified by proteomic analysis (Table S3) have been provided as supplementary materials. The raw mass spec files for proteomics analysis of S-glutathionylation were uploaded to MassIVE and can be accessed via the following link ftp://massive.ucsd.edu/MSV000085329/ The raw mass spectrometry files for global proteomic analysis were uploaded to MassIVE and can be accessed via the following link ftp://massive.ucsd.edu/MSV000084961/ The image files for ROS (Fig 2a and b) and senescence (Fig 4b and c) analyses have been included as source data files.

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
