## [Decision Letter]

**Acceptance summary:**

The manuscript details effects of the mitochondrially targeted peptide SS-31/elamipretide in the context of age-related mitochondrial dysfunction in aging mice. Similarities are drawn to effects seen in the mCAT in mice. The authors compare these effects on cardiac function in aged mice, relative to young mice and also in mCAT mice. This work demonstrates that SS-31 peptide improves age-dependent decline in cardiac function, partially through reduction in mitochondrial ROS and by mimicking effects seen in mCAT mice. The work provides comprehensive evidence that interventions targeting these mechanisms, later in life can reverse age-induced cardiac dysfunction.

**Decision letter after peer review:**

Thank you for submitting your article "Late-life restoration of mitochondrial function reverses cardiac dysfunction in old mice" for consideration by *eLife*. Your article has been reviewed by two peer reviewers, and the evaluation has been overseen by a Reviewing Editor and Jessica Tyler as the Senior Editor The following individual involved in review of your submission has agreed to reveal their identity: Pamela Boon Li Pun (Reviewer #2).

The reviewers have discussed the reviews with one another and the Reviewing Editor has drafted this decision to help you prepare a revised submission.

Summary:

In their manuscript, the authors present the ability of a mitochondrially targeted peptide SS-31/elamipretide to alleviate age-related mitochondrial dysfunction and thereby restore cardiac function, likely acting through reduction in oxidative damage. Similarities are drawn to benefits seen from over-expression of mCAT in old mice and, taken together, the conclusion is that application of the peptide can improve cardiac function through reduction in mitochondrial ROS and this supports their hypothesis that mitochondrial targeted therapeutics may be beneficial in reversing the effects of cardiac ageing. A significant amount of similar work has previously been published e.g. on the benefits of the peptide to improve mitochondrial dysfunction, protect against hypertensive cardiomyopathy, improve cerebrovascular endothelial function and thereby improve cognition in aged mice. However, there is novelty in the current work both in evaluating the effects on cardiac function in aged mice relative to young mice and in the deeper analysis of mechanism that is provided and in relating effects to those seen in the mCAT mice.

The work provides some evidence that delivery of interventions in later life can reverse age-induced cardiac dysfunction, although it is not entirely clear if this is purely reversal of age-dependent changes or general improvement (see below). The focus on late-life is valuable, although it would have been interesting to see if these benefits translate into increase lifespan or health span, if continued. The study is quite comprehensive, providing extensive data towards possible underlying mechanisms, related to metabolic and mitochondrial changes, redox signalling ROS and protein modifications.

Essential revisions:

1) There is a fundamental concern in that a similar improvement can be seen with SS-31 in young mice – which the authors show but do not discuss in Figure 1E. The authors do not present any other data on effects of SS-31 in young mice but, given the significant improvements in exercise duration, it is likely that benefits would be seen in other parameters in young mice as well (had these been measured / reported). Hence, benefits may not be seen exclusively in old mice or due to "reversal" of age-related cardiac function. Are these benefits of SS-31 on exercise in young mice from Figure 1E significant? Why are no other details on changes (markers) presented for young mice? The authors should present and carefully analyze any data that they have on this question and discuss its implications fully.

2) Gender differences in changes in mitochondria function and treatment effects have previously been reported. The Materials and methods section (subsection “Animals”, second paragraph) seem to indicate that the authors have studied both genders. If indeed so, the authors should present all such data and clarify if any gender differences were observed in this study. For example, data from female mice were used in Figure 1F. It would be useful to indicate as well whether the effects of SS-31 treatment persisted in male mice, as it did in female mice. It would also be useful to clarify in the legend for all figures and in the figures (panels) themselves, whether data from male and/or female mice is presented. In particular, authors should clarify in Figure 7 whether female mice (or male or a mixed-gender group) were used for the comparative SS-31 treatment group, since the mCAT mice were female (as indicated in the third paragraph of the subsection “Animals”).

3) The explanation of the statistics and experimental design should be clearer and carefully checked for consistency. For example, for measurement of protein S-glutathionylation; under Materials and methods the n=3 / group but in the figure text (Figure 4A) mentions n>6 / group.

4) Please also indicate the statistical analyses used for each figure in the legends and the exact n-numbers for each group rather than a range (especially since your range is very wide, e.g. Figure 1F n=3 to 7/group).

5) Just by eyeballing, it is very difficult to see how the authors achieve statistical significance, given the large standard deviation and low n-numbers e.g. Figure 1B, D, E and Figure 4B. A much clearer explanation of the statistical analyses needs to be given and raw data and the specific tests should be provided in each case.

6) I assume that for several of the studies presented, the same mice were followed (e.g. for Figure E and F, it looks like female mice were measured at baseline, post treatment and then even further followed until well after the end of treatment). Would it not make sense to take advantage of this fact in the statistical analysis by plotting / analyzing individual trajectories?

7) Why were young mice (Figure 1E) not followed up further (as in Figure 1F)? If such data has been collected it should be included and discussed.

8) It is heartening that late-life treatment was able to reverse cardiac dysfunction, yet it is also somewhat disappointing that the reversal was itself reversible once treatment was stopped. The authors have presented extensive changes in ROS levels, mitochondrial respiration, protein modifications, and metabolic profiling associated with late-life treatment. It would be desirable to present at least some data to indicate whether these changes were also reversible once treatment stopped.

9) Blots should be done in multiple replicates (if this was done, it is not indicated).

10) One of the key strengths of this study is the range and quality of techniques applied to the intervention study. However, to maximize the value of these data, it is important that all the raw data is made available in full and is annotated such that other can further or re-analyze it. The metabolomics data is provided in such a format but the some of the other data either require clearer annotation or are missing.

Examples of data that should be provided in full with appropriate annotation include:

• Full proteomics data on S-glutathionylation (Figure 4B).

• All images used for image-based analysis (ROS: Figure 2A and B).

• All images included in imaged-based analysis / statistics (Senescence: and Figure 4C and D).

---

## [Author Response]

Essential revisions:1) There is a fundamental concern in that a similar improvement can be seen with SS-31 in young mice – which the authors show but do not discuss in Figure 1E. The authors do not present any other data on effects of SS-31 in young mice but, given the significant improvements in exercise duration, it is likely that benefits would be seen in other parameters in young mice as well (had these been measured / reported). Hence, benefits may not be seen exclusively in old mice or due to "reversal" of age-related cardiac function. Are these benefits of SS-31 on exercise in young mice from Figure 1E significant? Why are no other details on changes (markers) presented for young mice? The authors should present and carefully analyze any data that they have on this question and discuss its implications fully.

We want to stress that in Figure 1E the trend for increased running time in young male mice treated with SS-31 mice was not statistically significant. Also, in a female cohort we previously tested, we also observed no significant improvement in treadmill running in young mice (Campbell et al., 2018).

Furthermore, we did not observe significant differences in the other parameters that we measured in young mice with and without SS-31 treatment, including exercise performance (Figure 1E), cardiac hypertrophy (Figure 1D) and metabolome (Figure 5—figure supplement 1). These results suggest that SS-31 is effective in aged hearts with pre-existing mitochondrial dysfunction but has little effects in young hearts with normal functioning mitochondria, as has been previously observed (Campbell et al., 2018; Siegel et al., 2013; Szeto, 2013). Therefore, we focused on the treatment effects in old mice and did not look at the remaining parameters in young SS-31. We have now added this in the Discussion section (subsection “Targeting mitochondrial oxidative stress in late-life reverses cardiac aging phenotypes”).

2) Gender differences in changes in mitochondria function and treatment effects have previously been reported. The Materials and methods section (subsection “Animals”, second paragraph) seem to indicate that the authors have studied both genders. If indeed so, the authors should present all such data and clarify if any gender differences were observed in this study. For example, data from female mice were used in Figure 1F. It would be useful to indicate as well whether the effects of SS-31 treatment persisted in male mice, as it did in female mice. It would also be useful to clarify in the legend for all figures and in the figures (panels) themselves, whether data from male and/or female mice is presented. In particular, authors should clarify in Figure 7 whether female mice (or male or a mixed-gender group) were used for the comparative SS-31 treatment group, since the mCAT mice were female (as indicated in the third paragraph of the subsection “Animals”).

We have clarified in the figure legends the sexes of mice used in each experiment.

We assessed the effect of SS-31 treatment on diastolic function in both male (Figure 1A) and female (Figure 1F) mice and observed similar improvement. We also observed similar benefits in exercise performance in male and female mice after 8-week SS-31 treatment (Figure 1E and Campbell et al., 2018). However, due to limitations in numbers of old mice available, not all parameters were tested in both sexes. Persistence of the SS-31 induced improvement was only followed in female mice but not male mice. We used female WT mice for AAV9-mCAT administration (Figure 7A, C, D and E), and female WT mice treated with SS-31 were used in Figure 6 for comparison of the effects of AAV9-mCAT and SS-31 on myofilament protein phosphorylation.

For Figure 7B, a mixed-sex group of mCAT transgenic mice and WT littermates were used to study the effect of SS-31 in combination with mCAT transgenic expression. We used a mixed-sex group because we only had a limited number of old mCAT transgenic mice available and because we had observed a similar diastolic function benefit from SS-31 treatment in male and female WT mice.

3) The explanation of the statistics and experimental design should be clearer and carefully checked for consistency. For example, for measurement of protein S-glutathionylation; under Materials and methods the n=3 / group but in the figure text (Figure 4A) mentions n>6 / group.

We apologize for the mistake in the Figure 4A legend. For protein S-glutathionylation analysis, n=3/group were used. We have corrected the mistake in the figure legend.

4) Please also indicate the statistical analyses used for each figure in the legends and the exact n-numbers for each group rather than a range (especially since your range is very wide, e.g. Figure 1F n=3 to 7/group).

Thank you for this suggestion. We have now included the statistical analyses used and specified the exact sample sizes in the legends.

5) Just by eyeballing, it is very difficult to see how the authors achieve statistical significance, given the large standard deviation and low n-numbers e.g. Figure 1B, D, E and Figure 4B. A much clearer explanation of the statistical analyses needs to be given and raw data and the specific tests should be provided in each case.

Graphs were previously plotted as mean +/- SD so the variations appeared larger than graphs that were plotted as mean +/- SEM. They are now plotted as mean +/- SEM with individual data points. We have clarified the specific statistical analyses used for each experiment in the legends.

6) I assume that for several of the studies presented, the same mice were followed (e.g. for Figure 1E and F, it looks like female mice were measured at baseline, post treatment and then even further followed until well after the end of treatment). Would it not make sense to take advantage of this fact in the statistical analysis by plotting / analyzing individual trajectories?

As suggested by the reviewer, we plotted the individual trajectories of cardiac function and exercise performance and included the results in Figure 1—figure supplements 1-3.

To take advantage of the individual trajectories in statistical analysis, we also performed linear regression analyses treating each mouse in a group as a separate data point. In agreement with the repeated measure ANOVA in Figure 1A and B, linear regression also suggested significant improvement in diastolic function and myocardial performance over time for old SS-31 treated but not old control mice. These statistical analyses are described in the legend of Figure 1—figure supplement 1.

7) Why were young mice (Figure 1E) not followed up further (as in Figure 1F)? If such data has been collected it should be included and discussed.

The effects of SS-31 on exercise performance (Figure 1E), cardiac hypertrophy (Figure 1D) and metabolome (Figure 5—figure supplement 1) were assessed in both young and old mice. We did not observe significant differences in these parameters in young mice after SS-31 treatment, and thus, there was no significant effect to follow. As noted above, and discussed in the text, SS-31 appears to be effective only when there is pre-existing mitochondrial dysfunction but has little effects on normal mitochondria in young mice. Therefore, we didn’t follow up on young mice in other experiments.

8) It is heartening that late-life treatment was able to reverse cardiac dysfunction, yet it is also somewhat disappointing that the reversal was itself reversible once treatment was stopped. The authors have presented extensive changes in ROS levels, mitochondrial respiration, protein modifications, and metabolic profiling associated with late-life treatment. It would be desirable to present at least some data to indicate whether these changes were also reversible once treatment stopped.

We agreed that it is a little disappointing that the benefit of SS-31 is not more persistent. One interesting observation is that the persistence of the functional benefit varied between individual mice, the individual trajectories is now included in Figure 1—figure supplement 3. We hypothesize that the persistence of the molecular changes e.g. oxidative damage and myofilament protein phosphorylation, also varied between individuals. Future studies will be required to follow up on how different molecular changes mediated by SS-31 persist after treatment cessation and determine the molecular mechanisms driving this individual variation and limiting SS-31 persistence. We have added this limitation to the Discussion section (subsection “SS-31 and mCAT expression differentially regulate phosphorylation of myofilament proteins to improve diastolic function”).

9) Blots should be done in multiple replicates (if this was done, it is not indicated).

Each immunoblotting was performed with at least 3 biological replicates (heart samples from 3 mice) per treatment/age group. We have now listed the exact numbers of biological replicates in the figure legends.

10) One of the key strengths of this study is the range and quality of techniques applied to the intervention study. However, to maximize the value of these data, it is important that all the raw data is made available in full and is annotated such that other can further or re-analyze it. The metabolomics data is provided in such a format but the some of the other data either require clearer annotation or are missing.Examples of data that should be provided in full with appropriate annotation include:• Full proteomics data on S-glutathionylation (Figure 4B).

Raw mass spec files for S-glutathionylation have been deposited raw data into MassIVE and can be accessed via the following link (included in the Materials and methods section): https://massive.ucsd.edu/ProteoSAFe/dataset.jsp?task=e8e88dd88dfc416d962d6cedf7af90d1

We are currently preparing a manuscript on the protein by protein analysis on SS-31 effects on S-glutathionylation. We have added this detail to the Discussion.

• All images used for image-based analysis (ROS: Figure 2A and B).

All image files for Figure 2A and B were included as supporting zip documents in the submission.

• All images included in imaged-based analysis / statistics (Senescence: and Figure 4C and D).

All image files for Figure 4C and D were included as supporting zip documents in the submission.